# Fugatto 1
## Foundational Generative Audio Transformer Opus 1

**NVIDIA**
Rafael Valle, Rohan Badlani, Zhifeng Kong, Sang-gil Lee, Arushi Goel, Sungwon Kim,
João Felipe Santos, Shuqi Dai, Siddharth Gururani, Aya AlJa'fari, Alexander H. Liu,
Kevin Shih, Ryan Prenger, Wei Ping, Chao-Han Huck Yang, Bryan Catanzaro
`rafaelvalle@nvidia.com`

## ABSTRACT

*Fugatto* is a versatile audio synthesis and transformation model capable of following free-form text instructions with optional audio inputs. While large language models (LLMs) trained with text on a simple next-token prediction objective can learn to infer instructions directly from the data, models trained solely on audio data lack this capacity. This is because audio data does not inherently contain the instructions that were used to generate it. To overcome this challenge, we introduce a specialized dataset generation approach optimized for producing a wide range of audio generation and transformation tasks, ensuring the data reveals meaningful relationships between audio and language. Another challenge lies in achieving compositional abilities – such as combining, interpolating between, or negating instructions – using data alone. To address it, we propose *ComposableART*, an inference-time technique that extends classifier-free guidance to compositional guidance. It enables the seamless and flexible composition of instructions, leading to highly customizable audio outputs outside the training distribution. Our evaluations across a diverse set of tasks demonstrate that *Fugatto* performs competitively with specialized models, while *ComposableART* enhances its sonic palette and control over synthesis. Most notably, we highlight emergent tasks and properties that surface in our framework's – sonic phenomena that transcend conventional audio generation – unlocking new creative possibilities. Demo Website.

## 1 INTRODUCTION

Recent research has sparked a strong debate between **specialist** and **generalist** models. While specialist models excel at specific tasks, they tend to be brittle, often struggling with changes in data distribution or task requirements. In contrast, generalist models eliminate the need for task-specific designs, can process diverse data, and scale effectively with increased compute and data. They also demonstrate emergent capabilities, enabling unsupervised task learning by leveraging broader datasets (Radford et al., 2019) and demonstrations (Brown et al., 2020). In this paper, we propose a strategy for developing a generalist audio synthesis and transformation model, called *Fugatto*, and an inference method for composing instructions through latent space manipulation, including from different models, called Composable Audio Representation Transformation (*ComposableART*).

Large language models (LLMs) have demonstrated impressive unsupervised multitask learning capabilities in the text domain (Radford et al., 2019), where instructions can be inferred from the data itself. However, such instructions are typically absent in the audio domain, making it difficult to generalize to unseen tasks without explicit guidance. Although models such as (Wang et al., 2024; Yang et al., 2023; Vyas et al., 2023) exist, they have several limitations enumerated in Table 1. In this panorama, dataset and instruction generation is necessary.

We employ a **multifaceted data and instruction generation strategy** that considerably expands the range of tasks of audio generation models. First, we use LLMs to generate and augment instructions and captions, providing *Fugatto* with instructions that are closer to free-form instructions (Goel et al., 2024; Doh et al., 2023). Second, we develop instructions that can be either absolute (e.g.,

"synthesize a happy voice") or relative (e.g., "increase the happiness of this voice"), enabling *Fugatto* to handle a wide array of dynamic tasks (OpenAI, 2024). Third, we leverage audio understanding models (Kong et al., 2024a; Gong et al., 2023) to create descriptions and synthetic captions for audio clips, enriching the dataset where annotations are sparse, allowing for better generalization and more accurate performance (Kong et al., 2024b). Fourth, we transmute existing datasets to uncover new relationships, enabling the creation of entirely new tasks without requiring additional raw data. Finally, we use audio processing tools to create new connections between text, audio, and their corresponding transformations.

By combining these approaches, we ensure that *Fugatto* has access to diverse and enriched datasets, allowing it to learn from various audio domains and contexts. This strategy enhances the model's task diversity and lays the groundwork for unsupervised multitask learning at scale, unlocking emergent abilities such as synthesizing entirely new sounds, such as "a saxophone barking".

While data is important, the ability to **compose, interpolate between, and negate instructions** is generally difficult to obtain through data alone. Negation data, for instance, is typically unavailable, and generating outputs that represent the composition or interpolation of instructions is equally challenging. Though this has been explored in the vision domain using Energy Based Models (EBMs) (Du et al., 2020) and EBMs in latent space (Nie et al., 2021), EBMs require training a new model per attribute, which can be cumbersome and impractical for a large number of attributes.

To overcome this limitation, we propose an inference-time technique called *ComposableART*, which is based on classifier-free guidance (CFG) (Ho & Salimans, 2022) and further expands on the dual classifier-free guidance in (Yang et al., 2024; Lee et al., 2024). We propose a generalized framework that leverages the weighted combination of vector fields between instructions, frame indices and models. This approach enables *Fugatto* to handle complex instruction-based operations, such as smoothly interpolating between instructions or negating specific instructions to exclude undesired effects. In contrast, models like (Vyas et al., 2023; Yang et al., 2023) rely on more rigid methods, often requiring external classifiers or manual intervention to achieve similar results.

In this paper, we present a detailed exploration of *Fugatto*'s dataset and instruction generation, training strategies, implementation improvements, and performance across a wide range of tasks. Through extensive evaluations and comparisons with specialist and generalist models, we establish *Fugatto* as a new benchmark for foundation models for audio synthesis and transformation. Similarly, we establish *ComposableART* as a highly desirable framework for compositional guidance that unlocks the full potential of score-based generative models. Our contributions include:

- We offer a comprehensive strategy for building a foundation model for audio generation and transformation given text and audio inputs, delivering strong performance across a wide range of tasks and providing a robust framework for both research and practical applications.

- We demonstrate how to enhance and create contextually rich audio and text datasets while generating flexible instructions with LLMs, enabling our community to replicate and adapt these techniques for their own models.

- We demonstrate how to perform composition, interpolation, and negation of instructions by extending CFG to compositional guidance, enabling better control over the model's outputs.

Table 1: Comparison of our proposed model *Fugatto* with other models.

| | *Fugatto* | AudioBox | NExT-GPT | UniAudio | AUDIT | VoiceLDM |
|---|---|---|---|---|---|---|
| Emergent properties | ✓ | ? | ? | ? | ? | ? |
| Large-scale data | ✓ | ✓ | ✓ | ✓ | ✗ | ✗ |
| Supports numerous tasks | ✓ | ✗ | ✗ | ✓ | ✗ | ✗ |
| Free-Form Instructions | ✓ | ✗ | ? | ? | ✓ | ✗ |
| Open-ended generation | ✓ | ✗ | ✗ | ✓ | ✗ | ✗ |
| Compositionality | ✓ | ✗ | ✗ | ✗ | ✗ | ✓ |
| Multi-Modal Inputs | ✓ | ✓ | ✓ | ✓ | ✓ | ✗ |

## 2 APPROACH

Our approach to audio generation given text and optional audio is similar to recent approaches in LLMs, focusing on large scale compute and datasets, followed by pre-training and fine-tuning stages. Our approach differs in two aspects: first, our dataset generation mechanism requires going beyond unsupervised next token prediciton (Section 2.1); second, we propose inference time techniques to control audio generation (Section 2.4). Appendix A.1 provides a list of tasks and instructions.

### 2.1 DATASET GENERATION

We aim to build a large and diverse dataset to capture demonstrations of a wide range of audio tasks and contexts. We emphasize that our ultimate goal is not to just excel on such tasks, but to drive, as a community, towards a future where unsupervised multitask learning emerges from data and model scale. Towards this goal, we propose a dataset generation strategy built on the five pillars below, and provide in Appendix A.1.1 a thorough description of each pillar.

**I – Generating Free-Form Instructions with LLMs:** Our strategy consists of prompting an LLM to create task-specific instruction generators, similar to Kocielnik et al. (2023). We prompt an LLM to create python methods that generate instructions of different lengths, and for different personas (standard, young-crowd, thirty-somethings, professional), given task-specific inputs including audio description, language, and others. Each persona has its own set of parts of speech.

**II – Generating Absolute and Relative Instructions:** Following GPT4-o's (OpenAI, 2024) ability to perform a relative change in speech, we aim to obtain an audio generation model that is able to follow instructions that are absolute or relative, such as "synthesize a happy voice" or "increase the happiness in this voice.". Given that such data is normally not available, we create it by transmuting existing datasets and leveraging audio processing tools. Thus, we can select one sample and create an absolute instruction for it, or select two samples and create a relative instruction. Similar to absolute instructions, relative instructions are produced by an LLM, that produces a python method that creates an instruction given the task, the attribute, and its modification.

**III – Generating Audio Descriptions with Audio Understanding Models:** We use audio understanding and classification models (Kong et al., 2024a) to produce synthetic captions for audio in the wild, following recent research (Leng et al., 2023; Kong et al., 2024b) showing that it is possible to drastically expand or improve text-to-audio models with synthetic captions. As such, we expand the strategies described in (Leng et al., 2023; Kong et al., 2024b) to generate high quality synthetic captions. For speech data, we implemented a prompt generation pipeline that automates the creation of natural language descriptions for voices. The pipeline converts speech attributes predicted by models – such as "gender", emotion, and speech quality – into detailed natural language descriptions using LLM-generated templates. These templates describe voices in various ways based on the speaker attributes, enhancing diversity by generating descriptions in multiple formats.

**IV – Creating New Tasks and Datasets by Transmuting Datasets:** We leverage implicit relationships between samples in a dataset to enable new tasks. Generally speaking, we look for datasets where one factor is held constant while other factors change. For example, we leverage emotional speech synthesis datasets with different renditions of the same text (Livingstone & Russo, 2018) by the same speaker to define a speech transformation task. Similarly, we leverage instrument synthesis datasets with different renditions of the same note (Engel et al., 2017) to define an instrument transformation task. We also leverage datasets that provide the individual parts of a sounds mixture (Rafii et al., 2017) to support tasks such as source separation, and audio generation conditioned on audio context and captions, possibly synthetic.

**V – Creating New Tasks and Datasets by Leveraging Audio Processing Tools:** We create synthetic paired data for speech and audio by using Praat (Boersma & Van Heuven, 2001) and Pedalboard (Spotify, 2024) to manipulate several speech and audio factors. For each factor, we apply controlled modifications, allowing us to generate speech and audio samples with specific alterations. With this strategy, we create speech-audio pairs for transformation tasks, like adjusting speech factors—e.g., "increase the F0 variance slightly and decrease the speech rate." or "add moderate reverb to this audio file". For each factor, we determine a practical range of adjustments and define increments that correspond to varying degrees of change, such as "slightly", "moderate", and "significant".

With these pillars established and leveraging open source datasets, we are able to build a large text and audio dataset with at least 20 million rows, not including on-the-fly modifications to captions, instructions and audio. Assuming each row refers to 10 seconds of audio, our dataset is comprised of at least 50,000 hours of audio. We provide a full list of datasets, tasks, and instructions in Appendices A.1.2, A.1.3 and A.1.4 respectively.

## 2.2 INSTRUCTION GENERATION

Our approach supports template-based and free-form instructions.

**Template-based Instructions:** In these instructions, the task is explicitly provided, followed by task-specific attributes, always in the same order, and with each attribute wrapped between start and end of attribute markers. We dynamically construct template-based instructions based on the task and the set of factors. Each factor is specified by its name and corresponding value, using the format `given {name}:{value}` followed by a closing HTML-like tag `</{factor}>`. The instruction always starts with the task, followed by its factors, and ends with `output:`. Additionally, a `given context` clause is appended at different locations, determined by the task at hand, when audio contexts are present. The structure of a template-based instruction is:

```
input:{task} given {factor}:{value}</{factor}>given context:<audio>output:
```

where {task} refers to the specific task, {factor} and {value} refers to the different factors and their respective values, and <audio> refers to the audio context provided. We provide examples of task and dataset specific instructions in Appendix A.1.4.

**Free-form Instructions:** We dynamically construct free-form instructions by using instruction generators introduced in Section 2.1. Unlike template-based instructions, where we know beforehand which part of the text refers to each audio context, free-form text makes it less straightforward to determine which words correspond to the audio. As such, in free-form instructions we simply append to the instruction `given context_k:<audio>` for each audio context, resulting in this structure:

```
input:{instruction} [given context:<audio_k>]output:
```

In order to promote simplicity and agility during development, we decided to use raw text instead of learnable tokens for `given factor` and `</factor>`. Following LLM practice, the full instruction and each audio are wrapped with learnable `<start of>` and `<end of>` tokens.

## 2.3 MODEL AND TRAINING

In this section we describe the text and audio representations used in our model, as well as the training objective and architecture. We provide a graphical depiction and details for hyperparameters, objective function, training stages and oversampling in A.2. In consonance with recent work in LLMs, we aim for a general model with weaker modeling assumptions, or inductive priors, versus a model with several hand-crafted or modality specific assumptions.

**Text and Audio representation:** The text representation is obtained by encoding the previously described instructions with a pre-trained language model held frozen during training. In this *Opus*, we use the *byT5* tokenizer free language model (Xue et al., 2022), which supports a large set of characters, including IPA. For now, the audio representation is a 22khz mel-spectrogram with 80 bins, which is subsumed by a relatively shallow learnable transformer encoder. The mel spectrogram is scaled to have approximately 0 mean and 1 standard deviation.

**Training Objective and Architecture:** We train our model with the Optimal Transport Conditional Flow Matching (OT-CFM) objective (Lipman et al., 2022; Tong et al., 2023), and use a T5-based (Raffel et al., 2020) Transformer (Vaswani, 2017) with Adaptive Layer Norm (Xu et al., 2019) as the parameterization of the vector field estimator. We replace the Transformer MLPs with kernel size 3 convolutions. After independently projecting the encoded text and audio to a shared embedding space, we time-wise concatenate the embedded audio and text. The model cross-attends to this representation, applying Adaptive Layer Norm (Xu et al., 2019) to them on every layer.

We observed that certain implementation choices yielded better training curves. Specifically, adaptive layer norm is completely computed in FP32, GELU uses approximate *tanh*, and the final layers are initialized to outputs zeros, which is approximately the mean of our scaled mel-distribution.

**Training Stages:** *Fugatto* training follows curriculum learning[1] (Bengio et al., 2009). We start with template-based instructions and a subset of tasks. Eventually, once we informally establish, by observing validation scores and listening to samples, that the model is able to follow template-based instructions, we proceed with an equal mixture of template-based and free-form instructions. Given that some tasks are underrepresented in the data, we find that oversampling leads to better results. Empirically, we find that starting sampling from a multinomial distribution with upsampling parameter $\beta = 0.25$, similar to Le et al., 2024, is sufficient. As training progresses, we adjust each dataset's weight according to validation scores on target tasks.

## 2.4 COMPOSABLE AUDIO REPRESENTATION TRANSFORMATION (*ComposableART*)

We extend Classifier Free Guidance(CFG) (Ho & Salimans, 2022) to support compositional guidance. Compositional guidance provides an OT-CFM model with the ability to independently control and generate (unseen) combinations of instructions and tasks, including with vector fields from different models (Karras et al., 2024). Compositional generation has been explored in Diffusion Models (Liu et al., 2023; Yang et al., 2024; Lee et al., 2024), for images and audio. To the best of our knowledge, we are the first to showcase novel ways of applying compositional guidance, expanding it to not just attributes, but also tasks, models and temporal composition of attributes.

**Compositional Guidance Method** Classifier Free Guidance(CFG), generally applied to diffusion models, combines the conditional and unconditional score estimates to obtain samples of higher quality and diversity pertaining to the condition. The following equation summarizes the application of CFG, where $\epsilon_\theta$ represents the score estimate and $\gamma$ represents the gradient scaling factor:

$$\epsilon_\theta(\mathbf{z}_\lambda, \mathbf{c}) = \epsilon_\theta(\mathbf{z}_\lambda, \mathbf{c}) + \gamma(\epsilon_\theta(\mathbf{z}_\lambda, \mathbf{c}) - \epsilon_\theta(\mathbf{z}_\lambda)) \tag{1}$$

We extend the Classifier Free Guidance framework to support the combination of vector fields across multiple instructions ($c_k$), multiple mel-frame indices (f) and multiple models ($\theta_m$). Let $v_{t,f}(c_k, \theta_m)$ be the vector field produced at flow-step t by model m, parameterized by $\theta_m$, given condition $c_k$ or $\varnothing$, for mel-frame f. Additionally, let $w_{k,f,m}$ be the flow-step invariant and user-determined scalar weight associated with $v_{t,f}(c_k, \theta_m)$. The equation for compositional guidance across instructions, frames, and models is defined as:

$$\tilde{v}_{t,f} = \sum_{k,m} w_{k,f,m}(v_{t,f}(c_k, \theta_m) + \gamma(v_{t,f}(c_k, \theta_m) - v_{t,f}(\varnothing, \theta_m))) \tag{2}$$

where $\gamma$ refers to the gradient scale parameter from CFG, and $\tilde{v}_{t,f}$ is the resulting composed vector field for flow step t and frame f. This follows similar conditional independence assumptions in (Nie et al., 2021; Liu et al., 2023) to support compositional guidance. We apply this compositional guidance at every step of the flow-matching inference procedure.

**Attribute/Event Composition:** An attribute is a simple input prompt belonging to a particular task, such as speech synthesis and audio event generation. A task can have multiple prompts or attributes as input. With compositional guidance, we can support unique unseen combinations of attributes. This gives the users an ability to create artistic combinations like simulating a scene with multiple-audio events, e.g. by composing 'thunder', 'rain' and 'wind' a storm can be achieved.

**Task Composition:** The model has been trained on many individual tasks, but it has not encountered combinations of tasks, such as speech synthesis alongside audio event generation. Using compositional guidance, we can enable the synthesis of unique, unseen task combinations, like generating speech with a specific audio event in the background.

**Model Composition:** The same technique can be extended to integrate distinct models. This is particularly useful when training domain-specific versions of *Fugatto*, each with its own parameters and datasets. This allows users to synthesize a "mixture of experts" sample that combines models trained on independent domains such as speech and audio events. Following (Karras et al., 2024), we use the velocities for each independent model, defined by parameters $\theta_m$.

---

[1]It simplifies development and supports incremental research, though not strictly necessary.

**Temporal Composition:** Instead of using the same scalar for every frame f, we assign a unique weight $w_{k,f,m}$ to each frame. This enables users to control the compositional output with arbitrary temporal curves (e.g., sigmoid or linear increase or decrease), while retaining the advantages of combining attributes, tasks, and models.

## 3 EXPERIMENTS

We present a comprehensive evaluation of *Fugatto* across multiple tasks to demonstrate its effectiveness and versatility. We begin with an ablation study, examining the impact of various design choices. Next, we evaluate *Fugatto*'s performance in audio synthesis and transformation tasks in speech, music and general sounds. Finally, we explore *Fugatto*'s emergent capabilities, and perform a thorough evaluation of our *ComposableART* method. Unless otherwise specified, we use template-based instructions, 2nd order Heun solver with 50 function evaluations, and task specific gradient scale $\gamma$.

### 3.1 ABLATIONS

We first analyze the impact of different $t$-sampling strategies in OT-CFM, comparing the traditional uniform sampling with others. Then, we examine the effect of model size on both loss metrics and emergent capabilities, including the ability to synthesize novel sounds not found in the training data—such as a "saxophone meowing" or "a person speaking while barking."

$t$**-sampling strategy:** In the OT-CFM framework, the timestep $t$ is typically sampled from a uniform distribution, $t \sim \mathcal{U}(0,1)$. However, recent discussions within the community have introduced conflicting strategies for this sampling process. Notably, Stable Audio's GitHub repository proposes sampling $t$ from a sigmoid-transformed normal distribution, $t \sim \text{sigmoid}(\mathcal{N}(0,1))$, thereby concentrating samples around $t = 0.5$. On the other hand, (Lovelace et al., 2023) advocate for increased sampling from values of $t$ closer to zero.

In our experiments, we observe that although Stable Audio's strategy provides a marginal improvement on TTA tasks, it renders the model unable to effectively attend to the transcript in text-to-speech (TTS) tasks – a critical requirement. We provide an explanation for this phenomenon in Appendix A.3.

> Training with $t \sim \mathcal{U}(0,1)$ is effective across all tasks.
> Training with $t \sim \text{sigmoid}(\mathcal{N}(0,1))$ significantly degrades TTS performance.

**Model capacity:** We evaluate how the learnable parameter count influences loss curves and emergent capabilities. We consider models with $0.8$ B, $1.4$ B params, and $2.5$ B parameters. Under a fixed data composition and sampling weights, we observe that increasing the parameter count from the smallest to the largest model not only improves validation losses but also delays overfitting. In Appendix A.2, we provide task-specific validation loss plots that showcase the expected decrease in validation loss as parameter count increases. Informally and consistent with findings in (Radford et al., 2019), we observe that the smaller model does not exhibit emergent abilities comparable to the larger models, particularly in their ability to synthesize novel sounds absent from the training data, such as "saxophone barking". We invite readers to evaluate samples in our supplementary materials.

> Emerging capabilities surface with sufficient model capacity and training data.

### 3.2 AUDIO SYNTHESIS

**Text-To-Voice Synthesis:** We evaluate in-context text-to-speech synthesis (TTS) and singing voice synthesis (SVS). For TTS, we follow the evaluation in Wang et al. (2023), using the same transcripts as Eskimez et al. (2024), to evaluate our model's ability to perform speech synthesis given a transcript and a speech sample from an unseen speaker. Following our training strategy, during evaluation we always provide the speaker's previous sentence when possible, otherwise a random sample different from the target. For SVS, we evaluate *Fugatto*'s ability to generate singing voice samples from instructions describing the desired lyrics and musical style without a backing track, for example: "Showcases a female singer with an interesting sound, conveys the message through american english lyrics, and infuses country influences throughout." The full list is available in Appendix A.4

Our zero-shot TTS results in Table 2a show that *Fugatto* has word error rates similar to ground truth, and is competitive with expert and generalist (Omni) models in terms of speaker similarity. Our SVS results in Table 2b shows high cosine similarity between CLAP embeddings (Wu et al., 2023) of the synthetic samples and the captions used to create it. We note that the higher WER in SVS likely comes from the higher difficulty for the generative model and the speech transcription model.

Table 2: *Fugatto* is comparable to generalist models and expert models on in-context TTS benchmarks. On SVS, it synthesizes samples with high CLAP-similarity and low WER relative to the task at hand.

(a) TTS on the LibriSpeech Test Clean benchmark in Wang et al. (2023)

| Model | Omni | WER $\downarrow$ | SIM-o $\uparrow$ | SIM-r $\uparrow$ |
|---|---|---|---|---|
| Ground Truth | | 2.20 | | |
| Vall-E (24khz) | ✗ | 5.90 | | 0.58 |
| Natural Speech 3 (16khz) | ✗ | 1.81 | 0.67 | 0.76 |
| AudioBox (16khz) | ✗ | 4.80 | 0.73 | |
| UniAudio (16khz) | ✓ | 2.00 | | 0.71 |
| *Fugatto* $\gamma = 2$ | ✓ | 2.66 | 0.60 | 0.61 |
| *Fugatto* $\gamma = 3$ | ✓ | 2.44 | 0.61 | 0.62 |

(b) SVS: WER and CLAP-Scores on a set of 10 music styles and 13 lyrics snippets from famous songs.

| Model | WER $\downarrow$ | CLAP $\uparrow$ |
|---|---|---|
| *Fugatto* $\gamma = 2$ | 71.90 | 0.49 |
| *Fugatto* $\gamma = 4$ | 19.54 | 0.45 |

**Text-To-Audio (TTA):** We showcase *Fugatto*'s performance on traditional TTA benchmarks that measure a model's ability to synthesize general sounds (AudioCAPS) and music (MusicCAPS) that follow instructions provided in text. We use the metrics (FD, FAD, and IS) and data splits (train, test) used in Kong et al. (2024b). Results in Table 3a and Table 3b shows that our model achieves strictly better scores than existing generalist models, while occasionally outperforming expert models.

Table 3: *Fugatto* outperforms generalists models and occasionally outperforms specialist models in TTA benchmarks on AudioCaps and MusicCaps.

(a) TTA on the AudioCaps benchmark in Kong et al., 2024b

| Model | Omni | FD $\downarrow$ | FAD $\downarrow$ | IS $\uparrow$ |
|---|---|---|---|---|
| VoiceLDM-Maudio | ✗ | | 2.50 | |
| AudioBox (16khz) | ✗ | 10.14 | 1.10 | 11.90 |
| NExT-GPT | ✓ | | 1.68 | |
| UniAudio | ✓ | | 3.12 | |
| *Fugatto* $\gamma = 1$ | ✓ | 16.73 | 1.36 | 9.72 |
| *Fugatto* $\gamma = 2$ | ✓ | 20.20 | 2.21 | 10.21 |

(b) TTA on the MusicCaps benchmark in Kong et al., 2024b

| Model | Omni | FD $\downarrow$ | FAD $\downarrow$ | IS $\uparrow$ |
|---|---|---|---|---|
| MusicGen (medium) | ✗ | 35.52 | 5.02 | 1.94 |
| AudioLDM-2-large | ✗ | 16.12 | 2.74 | 2.30 |
| Tango-AF&AC-FT-MC | ✗ | 21.84 | 1.99 | 2.21 |
| UniAudio* | ✓ | | 3.65 | |
| *Fugatto* $\gamma = 1$ | ✓ | 11.52 | 1.43 | 2.73 |
| *Fugatto* $\gamma = 2$ | ✓ | 13.18 | 1.93 | 2.97 |

### 3.3 AUDIO TRANSFORMATIONS

**Speech Enhancement:** We evaluate speech denoising and bandwidth extension tasks. Speech denoising evaluates the ability to extract speech from an additive mixture comprised of speech and noise. Bandwidth extension (sometimes referred to as "upsampling") evaluates the ability to recreate missing content from audio that is low passed filtered and downsampled not to include frequencies above a certain threshold frequency[2]. For speech denoising, we use the DNS-Noisy benchmark and traditional metrics PESQ and STOI described in Kong et al., 2023. For bandwidth extension, we use the VCTK benchmark and the traditional metric LSD described in Liu et al., 2024. In *Fugatto*, this task is interpreted as source separation, in contrast with the enhancement task that modifies the acoustic qualities of the target audio. Though *Fugatto* is comparable to specialist models in bandwidth extension, work remains to be done to close the gap in denoising.

**Speech Modulation:** In this task we evaluate our model's ability to transform a person's emotion in speech into another emotion, while preserving their speaker identity and the transcript. For this purpose, construct a train and test set based on the ESD dataset. We use open-source models to report emotion classification and correlation with Valence, Arousal and Dominance (VAD). We establish upper bounds by also computing scores on ground truth data. Our results in Table 5a show that our model is able to properly transform the emotion as well as the ground truth, it needs improvement on speaker similarity and word error rates.

---

[2]We use librosa after observing that torch.audio leaks frequencies above the threshold frequency.

Table 4: *Fugatto* is comparable to specialist models for speech denoising and upsampling.

(a) Speech Denoising on the DNS benchmark in Kong et al., 2023

| Model | Omni | PESQ$_{WB}$ ↑ | PESQ$_{NB}$ ↑ | STOI ↑ |
|---|---|---|---|---|
| Noisy dataset | | 1.59 | 2.16 | 91.60 |
| FullSubNet | ✗ | 2.90 | 3.37 | 96.40 |
| FAIR-Denoiser | ✗ | 2.66 | 3.23 | 96.60 |
| CleanUNet 2 | ✗ | 3.26 | 3.66 | 97.70 |
| *Fugatto* $\gamma = 0.1$ | ✓ | 2.77 | 3.32 | 95.70 |
| *Fugatto* $\gamma = 1$ | ✓ | 2.73 | 3.34 | 95.90 |

(b) Upsampling on the VCTK benchmark in Liu et al., 2024

| Model | Omni | LSD 4khz ↓ | LSD 8khz ↓ |
|---|---|---|---|
| Unprocessed (to 22khz) | | 2.74 | 1.84 |
| NuWave (to 22khz) | ✗ | 1.37 | 0.88 |
| NVSR (to 22khz) | ✗ | 1.49 | 1.37 |
| AudioSR (to 22khz) | ✗ | 1.25 | 1.08 |
| *Fugatto* $\gamma = 0.1$ | ✓ | 1.29 | 1.25 |
| *Fugatto* $\gamma = 1$ | ✓ | 1.38 | 1.34 |

**MIDI2Audio:** We evaluate our model's ability to convert midi to audio with the 250 simple 2-bar monophonic melodies from Pati et al., 2020. Note that *Fugatto* has never seen monophonic melodies during training, with the average number of stems present in training being 8. We use the error in pitch estimation on the ground truth MIDI notes as the upper-bound quality. We evaluate the L1 distance of the F0 contours extracted from the input rendered MIDI and the generated audio. During evaluation, we transpose the pitch contours to a common key. We provide examples in Appendix A.5.

Table 5: *Fugatto* performs well on novel tasks such as emotion conversion and MIDI2Aaudio.

(a) Speech modulation (Emotion Conversion): remarkably high Top-2 accuracy and pearson correlation $\rho$ between ground truth and synthetic samples on VAD. Low speaker similarity.

(b) MIDI2Audio: surprising zero-shot abilities on monophonic melody to audio.

| Model | WER | SIM-o | $\rho_{VAD}$ | Top-2 Acc. |
|---|---|---|---|---|
| Ground Truth | 7.42 | | | 0.62 |
| *Fugatto* $\gamma = 2$ | 24.17 | 0.19 | 0.68, 0.77, 0.77 | 0.59 |
| *Fugatto* $\gamma = 3$ | 21.96 | 0.21 | 0.68, 0.77, 0.77 | 0.62 |

| Model | L1 ↓ |
|---|---|
| F0 estimation error | 0.21 |
| *Fugatto* $\gamma = 1$ | 1.74 |
| *Fugatto* $\gamma = 2$ | 1.00 |

## 3.4 Emergent Capabilities and Sound Gallery

We consider an ability to be emergent if it is absent in smaller models but appears in larger ones (Wei et al., 2022; Radford et al., 2019), and *Fugatto* demonstrates *emergent sounds* and *emergent tasks*. In this section, we provide qualitative results through our demo page to highlight the model's emergent abilities and invite readers to a guided tour through our sound gallery, which showcases compelling examples of *Fugatto*'s artistic potential and emergent capabilities.

**Emergent Sounds:** *Fugatto* exhibits the ability to generate outputs that are not present in the training data itself. For instance, it can synthesize a cello that shouts with anger or a person that speaks and barks.

**Emergent Tasks:** Beyond generating novel sounds, *Fugatto* demonstrates the ability to perform tasks it was not explicitly trained for by combining tasks seen during training. For example, it can perform speech prompt conditioned singing voice synthesis or convert monophonic MIDI to a singing voice.

## 3.5 Compositionality

Compositionality enables users to combine attributes and tasks to generate novel input combinations not found in the training data, providing artistic and fine-grained control over the desired output. Since such novel combination rarely exists in the training data or the natural world, evaluating these samples is typically challenging. Furthermore, there are no established baselines and metrics for such sounds. Despite these challenges, we aim to provide a qualitative and quantitative evaluation of our proposed approach and invite readers to listen to compositional samples on our website.

### 3.5.1 Attribute/Event Composition

**Control intensity of each instruction (Weighted Combination)**: Compositional synthesis with instructions gives users a knob to control the intensity of each instruction. In order to evaluate this

ability, we create $\binom{10}{2}$ pairs of instructions by leveraging 10 event labels provided in Appendix A.6. Given a pair of events we can generate composite instructions through language or *ComposableART*:

Baseline linguistic composition example input:
```
input:  synthesize Event 1 and Event 2
```
Proposed *ComposableART* composition example inputs:
```
w1*v(input:  synthesize Event 1) + w2*v(input:  synthesize Event 2)
```

For each pair of events, we first generate samples using *ComposableART* with different convex combination of weights for each instruction, and using the linguistic baseline[3]. Then, for each method and generated audio, we compute the cosine similarity between the audio and text CLAP embeddings for each event in the event pair used to produce the audio, shown as Instruction 1 and 2 in Figure 1.

Figure 1 shows that as we increase the weight on Instruction 1, the occurrence of the event in the generated clip increases as evidenced by the CLAP scores. Reciprocally, as we decrease the weight on Instruction 2, the occurrence of the event in the generated clip decreases. In the linguistic baseline, we cannot control the weights of each attribute and their independent existence in synthesized samples is lower than the *ComposableART* samples at higher weights.

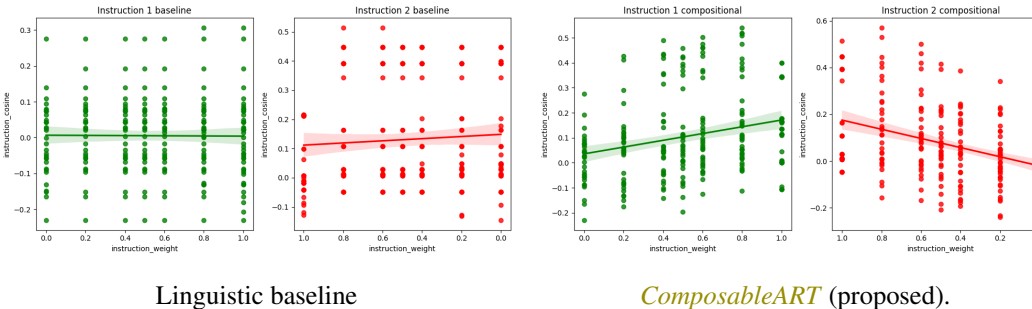

Linguistic baseline          *ComposableART* (proposed).

Figure 1: Comparison of CLAP scores between the Linguistic baseline and *ComposableART*'s composition of attributes with *Fugatto*. Instruction is equivalent to event.

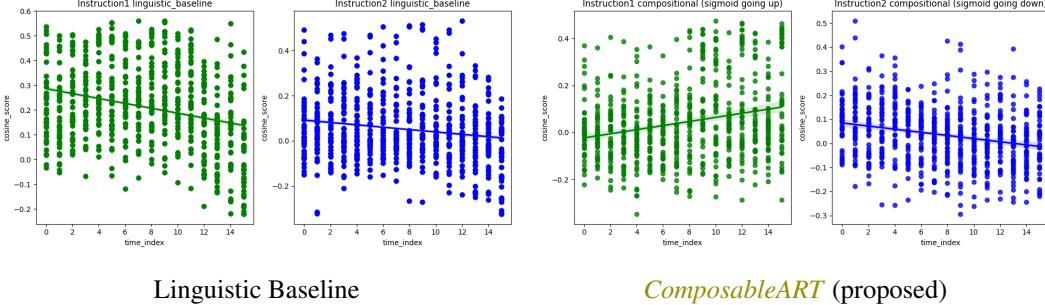

Linguistic Baseline          *ComposableART* (proposed)

Figure 2: Comparison of Baseline and *ComposableART* temporal guidance for instruction sequences.

**Negation of attributes**: With our approach, users can assign negative weights to attributes, producing negative velocity that steers the model away from the attribute. We use the previous $\binom{10}{2}$ event pairs to evaluate our approach alongside linguistic negation with the keyword 'NOT':

Baseline Linguistic Negation example input:
```
input:  synthesize Event 1 and NOT Event 2
```
*ComposableART* Negation example input:
```
v(input:  synthesize Event 1) - v(input:  synthesize Event 2)
```

Table 6 shows CLAP cosine similarity of the linguistic baseline against the proposed *ComposableART* method. It can be observed that while the positive event remains similar to the baseline, the negative event has a considerably lower cosine similarity than the baseline, indicating the effectiveness in

---

[3]In Figure 1, the diagonal fit, different weights, and the lack of samples in the linguistic baseline are an implementation and compute timeout byproduct given that the linguistic baseline does not support weights.

removal of the audio event using *ComposableART* as compared to the linguistic approach. Qualitatively, we also observe that this can be immensely useful in steering towards negative emotions in speech synthesis or swapping gender, a task which is not trivial.

**Interpolation of attributes**: *ComposableART* also supports interpolation of attributes by having the ability to independently control one attribute while keeping others. We evaluate our ability to control "pitch" as the interpolation variable by changing the weights on pitch and keeping "text" and "language" attributes fixed. Figure 3 showcases the expected decrease in fundamental frequency as we increase the weight on the "low pitch" attribute.

Table 6: CLAP Cosine Similarity of Attributes

| Instruction Type | CLAP Score |
|---|---|
| Linguistic baseline +ve Instruction | $0.02 \pm 0.02$ |
| Linguistic baseline −ve Instruction | $0.17 \pm 0.02$ |
| *ComposableART* +ve Instruction | $0.06 \pm 0.01$ |
| *ComposableART* −ve Instruction | $-0.04 \pm 0.02$ |

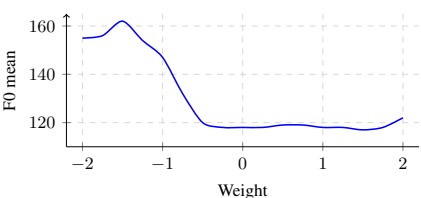

Figure 3: F0 mean given weight on the "low pitch" condition

### 3.5.2 TEMPORAL COMPOSITION

To evaluate the effectiveness of temporal guidance, we combine the previous set of audio events pairs over time. For *ComposableART*, we use a *sigmoid*-like curve to increase and decrease over time the weights on event 1 and event 2 respectively. For the equivalent linguistic baseline, we create the instruction "Event 2 followed by Event 1". Figure 2 showcases time-windowed CLAP scores, y-axis, for each approach and event across time, x-axis. We observe that the baseline is unable to establish the temporal trend as well as *ComposableART*, where we observe, as expected, a consistent increase in event 1 and consistent decrease in event 2.

### 3.5.3 TASK AND MODEL COMPOSITION

**Task Composition:** In our website, we provide qualitative samples where we compose a set of tasks involving "electronic music", "birds chirping", "dog barking", and "TTS". Our results show that the output conforms to the composition of such tasks using the proposed method.

**Model Composition:** In our website, we provide samples where we consider 2 different *Fugatto* models, one trained on speech datasets and the other trained on general sounds and audio events. We perform model composition to synthesize samples that contain both speech as well as audio events. This can be immensely useful in the future, where each domain specific model is an expert model and a combination of such high-quality experts can be used to synthesize compositional outputs without the need to train a monolithic large generative model.

## 4 DISCUSSION AND LIMITATIONS

We work towards a future where unsupervised multitask learning in audio synthesis and transformation emerges from data and model scale. Our proposed framework *ComposableART Fugatto* establishes our first step towards this direction, and in this step we become aware of our current limitations and challenges. For example, optimizing dataset sampling weights to drive performance on multiple benchmarks is a herculean task, and generative models for audio and video would certainly benefit from research similar to (Albalak et al., 2023; Chung et al., 2023; Xie et al., 2024). Along straighter paths, we plan to replace mels with a latent representation that better supports low frequencies and stereo. We believe this modification should be rather straight forward. Further work is necessary to establish the impact of data and free-form instructions on *Fugatto*'s emergent abilities, especially using language to combine tasks not jointly seen during training. Finally, *ComposableART* requires more analysis on the choice of weights, and how the norm of the vector field can be used for easier and more stable control.

REPRODUCIBILITY STATEMENT

We plan to release our synthetic captions, instructions and code to facilitate reproducible research.

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

# A APPENDIX

## A.1 DATASETS

In this section we provide information about our dataset generation strategy, tasks, instruction generators, and a full list of datasets created and used during training, including their sampling weights and task probabilities.

### A.1.1 APPROACH TO DATASET GENERATION

In this subsection we provide further descriptions of the procedures in each of the five pillars described in Section 2.1. Overall, the captions and instructions generation process is centered around leveraging LLMs to transform tags into richer descriptions, always being careful that the description relates to the audio content, and to create what we call *instruction generators*, i.e. python methods that create free-form instructions based on the synthetic descriptions and task at hand.

**I –** The python code snippet below is an LLM generated script that outputs free-form instructions for the task *reverse sound*, for example. Once such a script exists, it can be used to prompt an LLM to generate scripts for other tasks with increasing levels of complexity.

```python
class ReverseAudioInstructor:
    audio_references = {
        'standard': {
            'verbs': ['reverse', 'play backward', 'invert'],
            'gerunds': ['reversing', 'playing backward', 'inverting'],
            'contexts': ['audio', 'sound', 'recording', 'clip', 'track'],
            'asks': ['Can you', 'Please', 'Could you', 'I need you to'],
            'styles': ['completely', 'precisely', 'accurately'],
            'mentions': ['I provided', 'I sent', 'I attached']
        },
        ...
    }

    @staticmethod
    def generate_instruction(persona='standard'):
        ref = ReverseAudioInstructor.audio_references[persona]

        templates = [
            "{ask} {verb} the {context}.",
            "{verb} the {context} {style}.",
            "{ask} {verb} the {context} {mention}.",
            "Your task is to {verb} the {context}.",
            "We need the {context} {mention} to be {gerund}.",
            "{gerund} the {context} is the goal.",
            "Please focus on {gerund} the {context}.",
            "The objective is {gerund} the {context} {mention}.",
        ]

        template = random.choice(templates)

        instruction = template.format(
            ask=random.choice(ref['asks']),
            verb=random.choice(ref['verbs']),
            gerund=random.choice(ref['gerunds']),
            context=random.choice(ref['contexts']),
            style=random.choice(ref['styles']),
            mention=random.choice(ref['mentions'])
        )

        return instruction.capitalize()

if __name__ == '__main__':
    for persona in ReverseAudioInstructor.audio_references.keys():
        print(f"\n{persona.capitalize()} instructions:")
```

**II –** The python code snippet below is an LLM generated script that outputs free-form instructions for the task *speech modulation*, for example.. Note that the instructions refer to relative changes such as increase and decrease with different magnitudes such small or large changes. Once such a script exists, it can be used to prompt an LLM to generate scripts for other tasks with increasing levels of complexity.

```python
class SpeechModulationInstructor:
    instructions = {
        'scale_formant': {
            'increase': {
                'small': [
                    "add a touch more resonance",
                    "slightly enhance the formant frequencies",
                    ...
                ],
                ...
                'large': [
                    "dramatically enhance the resonance",
                    "massively boost the formant frequencies",
                    ...
                ]
            },
            'decrease': {
                'small': [
                    "tone down the resonance just a little",
                    "slightly reduce the formant frequencies",
                    ...
                ],
                ...
                'large': [
                    "dramatically reduce the resonance",
                    "massively lower the formant frequencies",
                    ...
                ]
            }
        },
        ...
        }
    }

    @staticmethod
    def get_instruction(modulation, direction, intensity):
        return random.choices(SpeechModulationInstructor.instructions[modulation][direction][intensity])[0]

    @staticmethod
    def combine_instructions(modulations):
        parts = []
        for modulation_i in modulations:
            modulation = modulation_i['modulation']
            direction = modulation_i['direction']
            intensity = modulation_i['intensity']
            instruction_part = SpeechModulationInstructor.get_instruction(modulation, direction, intensity)
            parts.append(instruction_part)
        instruction = ""
        if parts:
            # Combine parts into a single instruction
            if len(parts) > 1:
                combined_instruction = ", and ".join(parts[:-1]) + ", and " + parts[-1]
            else:
                combined_instruction = parts[0]
            instruction = "Let's " + combined_instruction + "."
        return instruction

if __name__ == '__main__':
    print("main")
```

**III -** Figure 4 provides a visual depiction of our speech captioning, dubbed Prompt-2-Voice (P2V) pipeline. We applied this approach to several open-source soeech datasets. Additionally, we incorporated existing speaker descriptions from datasets like PromptSpeech (Guo et al., 2023), which provide details on gender, pitch, volume, and speaking rate, and enriched them by adding emotion and quality information. This automation not only streamlines the process but also allows us to efficiently label in-the-wild datasets, significantly scaling the available training data wigh high quality captions.

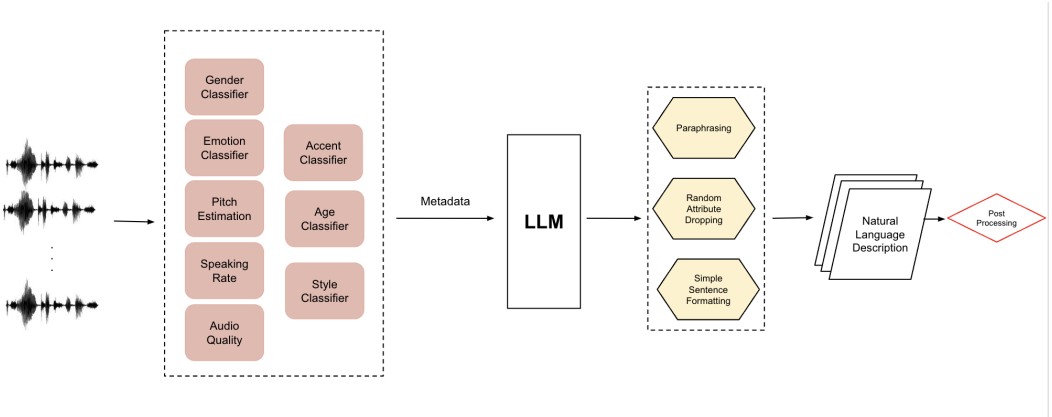

Figure 4: Synthetic caption generation pipeline for Prompt-to-Voice (P2V).

**V -** We use audio effects libraries to create synthetic paired data where some factors are held constant while others change. We use Pedalboard (Spotify, 2024), a Python library for audio effects, to apply different audio effects to sounds believed not to have effects. For each effect, there are a number of parameters that can be altered. We generate on the fly audio segments with the same effect but different levels of a single parameter while keeping the others fixed, so that we can not only create instructions with synthetic pairs of uneffected vs. effected data, but also synthetic pairs where we ask the model to increase or decrease the intensity of a given parameter (e.g. "increase the compression rate a little bit" or "reduce the room size for the reverb moderately"). For each parameter, we determined a reasonable range for the parameter, and increments that would correspond to "a little", "moderate", and "a lot" of that parameter.

A.1.2 LIST OF DATASETS

**Open Source Vocal Datasets**

Table 7: Open source vocal datasets. tts is Text-To-Speech and tta is Text-To-Audio (Audio from Caption)

| Dataset | Sampling Weights | Task and Probabilities |
| --- | --- | --- |
| AISHELL-3 | 2.46 | tts |
| CML-Dutch | 0.769 | tts |
| CML-French | 0.334 | tts |
| CML-German | 1.762 | tts |
| CML-Italian | 0.158 | tts |
| CML-Polish | 0.051 | tts |
| CML-Portuguese | 0.086 | tts |
| CML-Spanish | 0.512 | tts |
| common-accent-AccentClassification | 0.131 | tta |
| CREMA-D-EmotionClassification | 0.074 | tta |
| DAPS-Enhancement | 0.21 | enhancement paired 0.99 deenhancement paired 0.01 |
| DNS-Challenge-2020 | 14.247 | source separation |
| emov-db-EmotionClassification | 0.068 | tta |
| IEMOCAP-EmotionClassification | 0.059 | tta |
| jl-corpus-EmotionClassification | 0.024 | tta |
| LibriTTS-Clean-100 | 1.23 | tts |
| LibriTTS-Clean-360 | 4.31 | tts |
| LibriTTS-Other-500 | 0.643 | tts |
| LibriVox-English | 52.54 | tts |
| LibriVox-French | 0.909 | tts |
| LibriVox-German | 1.147 | tts |
| LibriVox-Italian | 0.114 | tts |
| LibriVox-Portuguese | 0.12 | tts |
| LibriVox-Spanish | 0.276 | tts |
| LIMMITS2024-* | 0.768 | tts 0.80 inpainting 0.10 upsampling 0.10 |
| MSP-PODCAST-Publish-1.9-EmotionClassification | 0.451 | tta |
| NonSpeech7k-EventClassification | 0.063 | tta |
| OMGEmotion-EmotionClassification | 0.017 | tta |
| ravdess-EmotionClassification | 0.014 | tta |
| SongDescriber-AudioCaptioning | 0.077 | tta |
| SONYC-UST-EventClassification | 0.279 | tta |
| tess-EmotionClassification | 0.028 | tta |
| VCTK-VoiceConversion | 2.89 | voice conversion paired vctk |
| VCTK-TTS | 0.137 | tts |
| VocalSound-VocalClassification | 0.155 | tta |

**Open Source Non-Vocal Datasets**

Table 8: Open Source Non-Vocal Datasets. tts is Text-To-Speech and tta is Text-To-Audio (tta)

| Dataset | Sampling Weights | Task and Probabilities |
|---|---|---|
| audiocaps-AudioCaptioning | 5.21 | tta 0.90
inpainting 0.10 |
| BBCSoundEffects-AudioDescription | 0.15 | tta |
| chime-home-EventClassification | 0.05 | tta |
| Clotho-AQA-EventClassification | 0.01 | tta |
| Clotho-AQA singlelabel-EventClassification | 0.07 | tta |
| Clotho-v2-AudioCaptioning | 0.19 | tta |
| CochlScene-SceneClassification | 0.61 | tta |
| Epidemic sound-AudioCaptioning | 0.41 | tta |
| ESC-50 | 1.12 | tta 0.90
inpainting 0.10 |
| FMA-GenreClassification | 1.04 | tta |
| FSD50k-EventClassification | 0.41 | tta |
| GTZAN-GenreClassification | 0.01 | tta |
| LP-MusicCaps-MC-AudioCaptioning | 0.07 | tta |
| LP-MusicCaps-MSD-0 | 8.56 | tta 0.90
inpainting 0.10 |
| LP-MusicCaps-MSD-1 | 8.62 | tta 0.90
inpainting 0.10 |
| LP-MusicCaps-MSD-AudioCaptioning | 11.82 | tta |
| LP-MusicCaps-MTT-AudioCaptioning | 0.47 | tta |
| MACS-AudioCaptioning | 0.17 | tta |
| Maestro | 0.08 | tta 0.90
upsampling 0.10 |
| Medley-solos-DB | 0.19 | tta 0.90
upsampling 0.10 |
| MSD | 12.38 | tta 0.80
upsampling 0.10
inpainting 0.10 |
| MTG-Jamendo | 2.16 | tta 0.90
inpainting 0.10 |
| musdbhq-InstrClassification | 0.10 | tta |
| MusicCaps-AudioCaptioning | 0.54 | tta 0.90
inpainting 0.10 |
| NonSpeech7k-EventClassification | 0.06 | tta |
| NSynth-MIR | 2.31 | tta |
| SongDescriber-AudioCaptioning | 0.08 | tta |
| SONYC-UST-EventClassification | 0.28 | tta |
| SoundDescs-AudioDescription | 0.23 | tta |
| SoundVE-Caps | 37.00 | tta |
| UrbanSound8K-EventClassification | 0.09 | tta |
| WavText5K-AudioCaptioning | 0.04 | tta |
| WavText5K-Tagging | 0.02 | tta |

**New Datasets generated from Open Source Data**

Table 9: New datasets generated from open source data. tts is Text-To-Speech, tta is Text-To-Audio (Audio from Caption), and P2V refers to prompt to voice, in which LLMs were used to create full form captions given speech attributes. The -AF suffix indicates synthetic captions generated with Audio Flamingo.

| Dataset | Sampling Weights | Task and Probabilities |
|---|---|---|
| audiocaps-AudioCaptioning-AF | 5.79 | tta 0.90 |
| | | inpainting 0.10 |
| AudioSet-AF | 37.24 | tta 0.80 |
| | | inpainting 0.05 |
| | | inpainting random mask 0.05 |
| | | upsampling 0.09 |
| | | downsampling 0.01 |
| AISHELL-3-AddRemove-Sound-Effects | 3.23 | add sound effects 0.50 |
| | | remove sound effects 0.50 |
| AISHELL-3-SoundEffectsModulation | 3.84 | sound effects modulation |
| AISHELL-3-SpeechModulationPraat | 3.84 | speech modulation praat |
| CLAP freesound-AF | 2.82 | tta |
| CREMA-D-P2V | 0.62 | tts 0.80 |
| | | inpainting 0.10 |
| | | upsampling 0.10 |
| EGFxSet-AddSoundEffects | 0.00 | add sound effects paired |
| EGFxSet-RemoveSoundEffects | 0.00 | remove sound effects paired |
| emov-db-EmotionClassification | 0.07 | tta |
| ESD-ChangeEmotion | 1.90 | speech modulation paired |
| ESD-ENGLISH-P2V | 0.93 | tts 0.80 |
| | | inpainting 0.10 |
| | | upsampling 0.10 |
| ESD-MANDARIN-P2V | 0.63 | tts 0.80 |
| | | inpainting 0.10 |
| | | upsampling 0.10 |
| IEMOCAP-EmotionClassification | 0.06 | tta |
| jl-corpus-EmotionClassification | 0.02 | tta |
| LibriTTS-Clean-100-Add-Remove-Sound-Effects | 7.85 | add sound effects 0.50 |
| | | remove sound effects 0.50 |
| LibriTTS-Clean-100-SoundEffectsModulation | 0.06 | sound effects modulation |
| LibriTTS-Clean-100-SpeechModulationPraat | 9.34 | speech modulation praat |
| LibriTTS-Clean-100-Enhancement | 0.85 | enhancement paired 0.99 |
| | | deenhancement paired 0.01 |
| LibriTTS-Clean-360-Enhancement | 2.71 | enhancement paired 0.99 |
| | | deenhancement paired 0.01 |
| LibriTTS-Other-500-Enhancement | 6.11 | enhancement paired |
| JL-CORPUS-P2V | 0.62 | tts 0.80 |
| | | inpainting 0.10 |
| | | upsampling 0.10 |
| LMD-Aligned | 1.17 | midi2audio |
| musdbhq-InstrClassification | 0.10 | tta |
| musdbhq-add-sound | 3.81 | add sound 0.50 |
| | | add sound to mixture 0.50 |
| musdbhq-singing | 0.86 | singing voice synthesis |
| musdbhq-singing-aggregated | 0.86 | singing voice synthesis |
| musdbhq-source-separation | 3.81 | source separation 0.50 |
| | | remove sound from mixture 0.50 |
| NonSpeech7k-EventClassification | 0.06 | tta |
| NSynth-MIR | 2.31 | tta |
| OMGEmotion-EmotionClassification | 0.02 | tta |
| ravdess-EmotionClassification | 0.01 | tta |
| RAVDESS-CreateVariation | 0.02 | speech modulation paired |
| RAVDESS-ChangeIntensity | 0.15 | speech modulation paired |
| RAVDESS-ChangeEmotion | 0.22 | speech modulation paired |
| RAVDESS-P2V | 0.62 | tts 0.80 |
| | | inpainting 0.10 |
| | | upsampling 0.10 |
| ExpressiveSinger-P2V | 2.84 | singing voice synthesis nolanguage |
| tess-EmotionClassification | 0.03 | tta |
| VGG-AF | 0.92 | tta |
| YODAS [English] (subset + additions) | 27.23 | tts 0.80 |
| | | inpainting 0.05 |
| | | inpainting random mask 0.05 |
| | | upsampling 0.04 |
| | | reverse sound 0.04 |
| | | downsampling 0.01 |
| WavCaps-AF | 7.83 | tta |

Below we provide descriptions for the suffixes associated with our generated datasets presented in Table 7, Table 8, and Table 9.

**–AF:** Refers to generating synthetic captions with the strategy described in (Kong et al., 2024b). In summary, the strategy consists of using an audio understanding model, here Audio Flamingo Chat, to caption sounds in the wild, and then filtering out synthetic captions based on the CLAP cosine similarity between the audio and the synthetic caption. In this paper, we used this strategy to create synthetic captions for AudioCaps, AudioSet, CLAP Freesound, and WavCaps.

**–{Add, Remove} Sound Effects:** These refer to applying, on the fly, audio effects modifications to existing audio data to create paired data that describe tasks related to adding and removing sound effects. We focus on speech, which we assume has the least amount of audio effects applied to it, especially when compared to music data. In this iteration, the audio modulations are performed with (Spotify, 2024) and include a large list of effects such as chorus, reverb, distortion, amongst others.

**–{Add, Remove}:** This refers to splitting an audio mixture into separate audio stems and creating manifests that add one track given another track. In this *opus*, we have not explored creating artificial mixtures by adding random waveforms but imagine this strategy can yield good results.

**–P2V:** P2V, or prompt-to-voice, is a task that enables control of speech synthesis through textual prompts, allowing for the description of speaker characteristics when an appropriate audio prompt that matches the desired persona is unavailable. The strategy has several components. First, we extract and curate all possible tags from existing metadata. For example, the expressive singer (Dai et al., 2024) dataset includes, implicitly and explicitly, information about vocal range, accent, language, style and others. Then, following the strategy in Section 2.1 we first prompt LLMs to produce long form descriptions of each attribute, then we prompt an LLM to create a instruction generator that combines the modified attributes in different ways.

**–Singing:** refers to leveraging music stems dataset with vocal tracks to create singing datasets. As usual, we leverage all the metadata available, combined with transcriptions obtained from speech transcription models and song captions obtained by prompting LLMs. In cases where an accompaniment or backing track is available, we associate the timestamps on each lyrics snippet with the respective accompaniment. Once the metadata types are available, we prompt an LLM to create a python method that produces instructions given the metadata for a singing voice synthesis task with or without accompaniment.

**–Singing-Aggregated:** -Singing-Aggregated adds to -Singing the combination of adjacent sentences, given criteria such as max gap between sentences and max sentence length.

**–Source Separation and Add To Mixture:** refers to leveraging existing In this iteration, assuming doing such would produce samples undesirable out-of-distribution samples, we do not create mixtures by combining random samples together. Instead, we leverage mixtures that are already exist.

**–Speech Modulation:** refers to applying speech modulations to existing speech data to create paired data that describe a task in which a relative change is being applied to an attribute of speech, e.g. "*Increase* the speaking rate in this sample". In this iteration, the speech modulations are performed with Praat (Boersma & Van Heuven, 2001) and we focus on transformations such as formant scaling, F0 mean scaling (equivalent to transposition in music), F0 variance scaling (equivalent to flattening or expanding the F0 contour), and speaking rate scaling (equivalent to making a person speak faster or slower). Due to artifacts that can be introduced in the audio as a result of the modulation, we limit the scaling values to ranges that we find acceptable in terms of audio quality.

**–Speech Modulation Paired:** This refers to leveraging available paired data to create new tasks. For example, an emotional speech dataset with paired data can be used to enable an emotion transformation tasks. The procedure is straight forward and consists of first grouping samples by speaker and transcript, then creating pairs that establish relationships between two samples. For example, given a speaker and transcript, two samples with different emotion can be used to define a "convert emotion task", two samples with the same emotion and intensity can be used to create a "create a variation of this speech", and two samples with the same emotion but different intensity can be used to define a "increase the intensity in the emotion task". Once the pairs and new tasks are defined, we prompt an LLM to create a script that takes in the attributes and tasks to generate task-specific instructions.

–**Sound Effects Modulation:** refers to applying sound effects modulations to existing data to create paired data for relative change in audio effects in a file. In this iteration, the speech modulations are performed with Praat (Boersma & Van Heuven, 2001) and focus on formant scaling, F0 mean scaling, F0 variance scaling and speaking rate scaling. We limit the values to the range below

### A.1.3 TASKS

Below we provide the list of tasks that are known to be supported by our model, and the respective minimal instruction for that task. Please note that we provide minimal instructions due to layout reasons, not due to model limitations.

| Task | Minimal Instruction |
|---|---|
| **Add Sounds to Instrument Track** | |
| using audio example | add drums like this `<audio>` to this guitar track `<audio>` |
| using text description | add rock drums to this guitar track `<audio>` |
| **Add Sounds to Sound Mixture** | |
| using audio example | add guitar like this `<audio>` to this backing track `<audio>` |
| using text description | add guitar to this backing track `<audio>` |
| **Apply Sound Effects** | add long reverb to this audio `<audio>` |
| **Copy Audio** | copy this `<audio>` |
| **Downsample Audio** | downsample this to 16kHz `<audio>` |
| **Enhance Audio** | enhance this sound `<audio>` |
| **Generate Audio From Captions** | generate a saxophone barking |
| **Generate Music From Captions** | generate a calm sound track |
| **Generate Speech From Captions** | generate an angry voice |
| **Inpaint Audio (Continuation)** | inpaint this sound `<audio>` |
| **MIDI2Audio** | |
| using audio example | turn this MIDI track `<audio>` into natural audio. |
| using text description | turn this MIDI track `<audio>` into a heavy metal track. |
| **Remove Sound Effects** | remove sound effects `<audio>` |
| **Remove Sound from Sound Mixture** | |
| from audio examples | remove this `<audio>` from these sounds |
| from text captions | remove the piano from this music track `<audio>` |
| **Reverse Sound** | reverse this sound |
| **Singing-Voice Synthesis** | |
| given Speech Captions | sing this 'At last my love' with a male voice |
| given Language | sing this 'At last my love' in English |
| given Accent | sing this 'At last my love' in English with an Italian accent |
| **Separate Audio into Sources** | |
| given audio examples | give me this `<audio>` from this music track `<audio>` |
| given text description | give me the piano from this music track `<audio>` |
| **Sound Effects Modulation** | (Increase and Decrease Attributes) |
| Chorus | increase the chorus a bit `<audio>` |
| Compressor | decrease the compressor threshold `<audio>` |
| Delay | decrease the delay time `<audio>` |
| Distortion | increase the distortion `<audio>` |
| Limiter | make the limiter less strong `<audio>` |
| Phaser | increase the phaser speed `<audio>` |
| Reverb | increase the reverb room size in this `<audio>` |
| **Speech Modulation (Paired)** | |
| Voice Conversion | convert from this `<audio>` to this `<audio>` |
| Accent Conversion | convert from British English to American English |
| Emotion Conversion | make this calm sample `<audio>` sound angry |
| Speech Variation | give me a different take on this voice `<audio>` |
| **Speech Modulation (Praat)** | |
| Scale Speaking Rate | increase the speaking rate `<audio>` |
| Scale F0 Mean | increase the average pitch `<audio>` |
| Scale F0 Variance | increase the variance in pitch `<audio>` |
| Scale Formants | scale the formants here `<audio>` |
| **Text-To-Speech** | |
| from Speech Prompt | say this 'May the force!' given this voice `<audio>` |
| from Speech Captions | say this 'May the force!' with a male voice |
| from Language | say this 'May the force!' in English |
| from Accent | say this 'May the force!' in English with an Italian accent |
| **Upsample Audio** | upsample this sound `<audio>` |

### A.1.4    INSTRUCTIONS

After an informal evaluation, we concluded that, for our purpose, Claude Sonnet is better than GPT4-o at producing instructions and following prompts. Below we provide examples of template-based and free-form instructions for a handful of tasks and datasets. For clarity, we remove the redundant 'input:' and 'output:' parts from all instructions.

AISHELL-3-AddRemove-Sound-Effects
```
Eradicating the audio from the provided audio material with the
Distortion, and Phaser effect is your task.  Focus on eradicating
it systematically.
```

AISHELL-3-SoundEffectsModulation
```
We need to minimize the delay time.</caption>
```

AISHELL-3-SpeechModulationPraat
```
Let's subtly enhance the pitch for a brighter sound, and
dramatically slow down the speech for a very relaxed
delivery.</caption>
```

audiocaps-AudioCaptioning
```
synthesize A consistent, loud mechanical motor</caption>
```

AudioSetFullwoAudioMusicCaps-EventClassification
```
synthesize This is a sound of Speech</caption>
```

AudioSet-AF
```
Yo, mind fill in the missing bits in this tune?  Please do this
smoothly.]
```

CREMA-D-P2V
```
Manifest a voice reproduction in American English verbalizing "The
airplane is almost full.", with the timbre is often monotone and
lacks energy, and itś like a middle-aged who is not Hispanic.
```

LibriTTS-Clean-100
```
Stitch up a spiel in en declaring ""But there was a passenger
dropped off for you-a little girl.", with a female speaker with
a moderate pitch and intonation delivers her words quite rapidly
in a confined, clear acoustic environment.
```

RAVDESS-ChangeEmotion
```
modulate I want to switch from angry to fearful.</caption> given
example:
```

## A.2 MODEL AND TRAINING

**Model visualization:** Figure 5 provides a visual description of *Fugatto*'s architecture and input handling and Algorithm 1 provides a pseudo-algorithm for the optimal-transport conditional flow matching loss. Essentially, it minimizes the mean squared error between the estimator's prediction and a linear interpolation between the data and the Gaussian noise sample used to condition the model on $x_t$, scaled by $(1 - \sigma)$, where $\sigma$ is a small enough value.

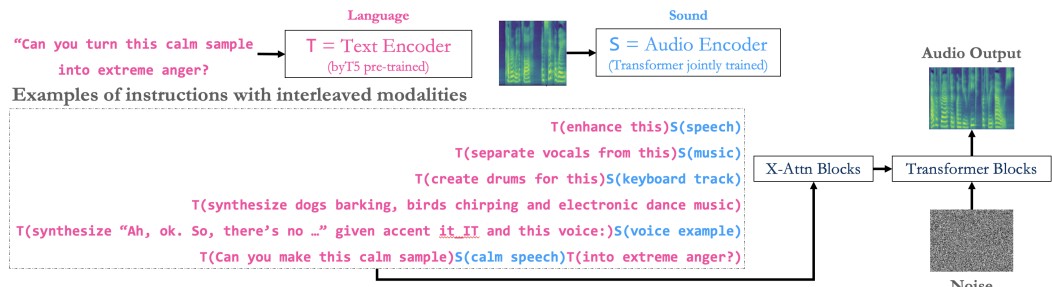

Figure 5: A description of *Fugatto*'s architecture and input handling.

---

**Algorithm 1** Optimal Transport Conditional Flow Matching Loss Pseudo-Algorithm

---

1: Sample $\mathbf{x}_1 \sim \mathcal{X}$, where $\mathcal{X}$ is the data distribution
2: $\sigma \leftarrow 0.01$
3: $\mathbf{x}_0 \leftarrow \texttt{randn\_like}(\mathbf{x}_1)$
4: $\mathbf{x}_t \leftarrow (1 - (1 - \sigma) \cdot t) \cdot \mathbf{x}_0 + t \cdot \mathbf{x}_1$
5: $\mathbf{v}_t \leftarrow estimator_\theta(\mathbf{x}_t, t; )$
6: $\mathbf{u}_t \leftarrow \mathbf{x}_1 - \mathbf{x}_0 \cdot (1 - \sigma)$
7: $\text{loss} \leftarrow \texttt{mse}(\mathbf{v}_t, \mathbf{u}_t)$

---

**Optimal Transport Conditional Flow Matching Pseudo-Algorithm:**

**Model hyperparameters:** We provide a list of *Fugatto*'s hyperparameters in Table 10.

Table 10: Model Hyperparameters for the main *Fugatto* evaluated in this paper.

| Hyperparameter | Value |
|---|---|
| t_schedule | uniform |
| n_mel_channels | 80 |
| n_hidden | 1536 |
| sigma | 0.01 |
| text_encoder_config.name | google/byt5-large |
| text_encoder_config.scale | 1.0 |
| text_encoder_config.n_hidden | 1536 |
| mel_encoding_strategy | separate |
| mel_encoder_config.is_causal | false |
| mel_encoder_config.pos_emb.name | rope |
| mel_encoder_config.pos_emb.base | 16384 |
| mel_encoder_config.use_flash_attention | true |
| mel_encoder_config.deterministic | false |
| mel_encoder_config.n_layers | 3 |
| mel_encoder_config.p_dropout | 0.1 |
| mel_encoder_config.p_dropout_out | 0.0 |
| mel_encoder_config.n_heads | 16 |
| mel_encoder_config.has_xattn | false |
| mel_encoder_config.apply_norm_to_cond | false |
| mel_encoder_config.layer_norm_method | pre |
| mel_encoder_config.kernel_size | 3 |
| mel_encoder_config.use_layer_scale | true |
| mel_encoder_config.layer_scale_init | 0.1 |
| mel_encoder_config.layer_scale_decay | 0.95 |
| mel_encoder_config.d_model | 1536 |
| decoder_config.d_time | 128 |
| decoder_config.transformer_hparams.is_causal | false |
| decoder_config.transformer_hparams.pos_emb.name | rope |
| decoder_config.transformer_hparams.pos_emb.base | 16384 |
| decoder_config.transformer_hparams.use_flash_attention | true |
| decoder_config.transformer_hparams.deterministic | false |
| decoder_config.transformer_hparams.n_layers | 24 |
| decoder_config.transformer_hparams.p_dropout | 0.1 |
| decoder_config.transformer_hparams.n_heads | 16 |
| decoder_config.transformer_hparams.has_xattn | true |
| decoder_config.transformer_hparams.kernel_size | 3 |
| decoder_config.transformer_hparams.context_xattn.n_heads | 16 |
| decoder_config.transformer_hparams.context_xattn.d_heads | 1536 |
| decoder_config.d_data | 80 |
| decoder_config.d_model | 1536 |

**Training and Inference** During the first phase, *Fugatto* is trained on at least 32 NVIDIA A100 GPU for approximately 1M iterations with template-based instructions and a subset of tasks. During the second phase, we restart the optimizer and train for approximately 250k iterations, sampling from template-based and free-form instructions uniformly and adding all tasks. We use the AdamW optimizer (Loshchilov & Hutter, 2019) with a learning rate of 1e-4, annealing the learning rate to 1e-6 during the second phase. A G2P model (Bernard & Titeux, 2021) pre-processes the text into the International Phonetic Alphabet (IPA) format. During inference, we generate mel-spectrograms using 50 function evaluations, 100 in practice, with Heun's Solver and task-specific guidance scale $\gamma$. Mel-spectrogram to waveform conversion is performed using the pre-trained universal BigVGAN V2 vocoder, available in the BigVGAN (Lee et al., 2023) repository[4].

---

[4]BigVGAN: https://github.com/nvidia/bigvgan

## A.3 ABLATIONS

**t-sampling** We draw inspiration from (Shen et al., 2018; Valle et al., 2020), where, due to optimization issues, the model would be stuck in a local minima and not make use of the text conditioning variable, rendering the model useless during inference. We believe the same is true with $t \sim \text{sigmoid}(\mathcal{N}(0, 1))$, and that with such a distribution the model is stuck in a local minima and does not leverage the text to minimize the loss.

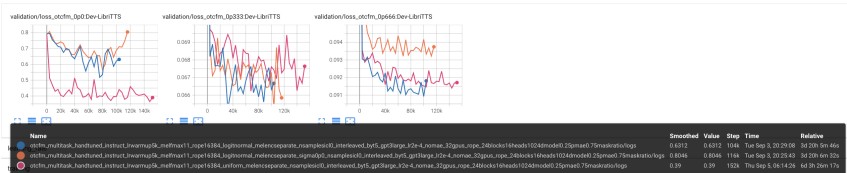

Figure 6: A description of the image.

**Model capacity:** Exponentially smoothed validation loss per task at different $t$-values from smaller models with $0.8$ B, $1.4$ B params, to larger models with $2.5$ B parameters.

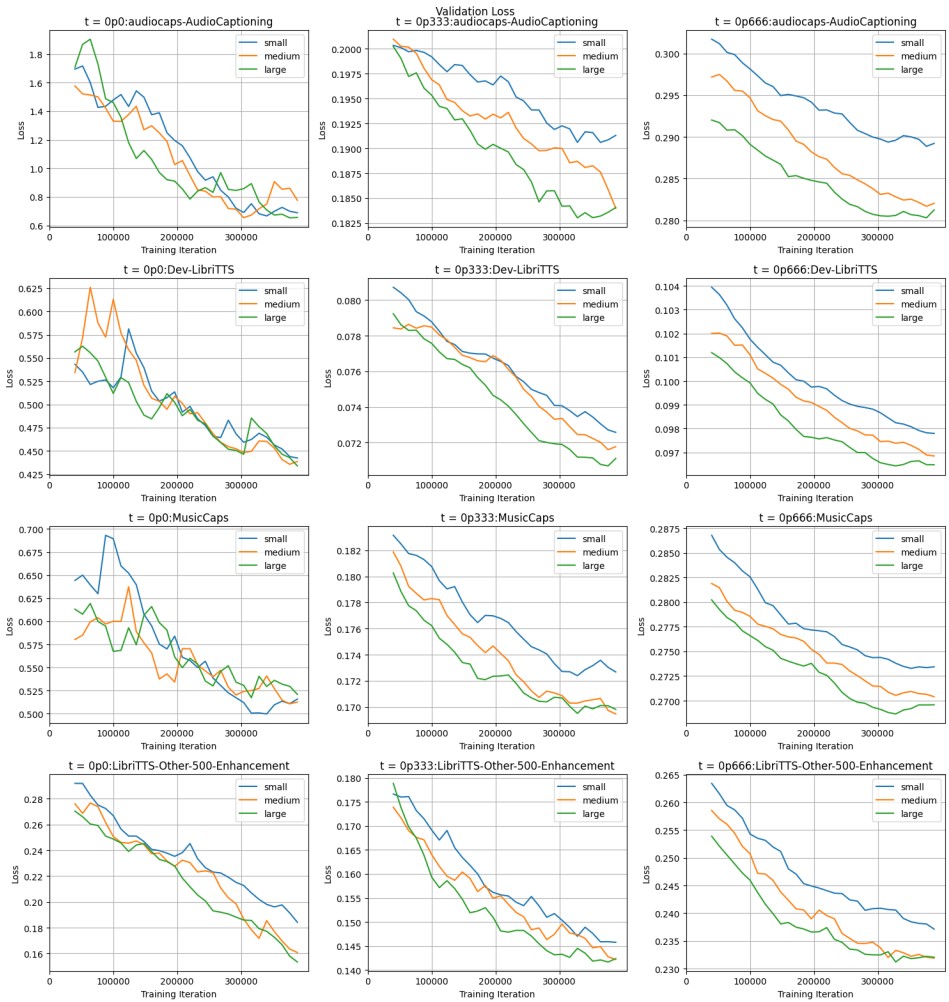

Figure 7: Validation scores for different *Fugatto* sizes on 3 benchmarks at 3 different $t$-values.

### A.4 SINGING VOICE SYNTHESIS MUSIC STYLES AND LYRICS

The Singing-Voice-Synthesis experiments in Section 3.2 evaluates all combinations between the 13 lyrics snippets and 10 music styles below. For each combination, we use the singing-voice-synthesis instruction generator to create instructions such as: "Showcases a female singer with an interesting sound, conveys the message through american english lyrics, and infuses country influences throughout."

This is the set of 13 lyrics snippets used during evaluation:

```
"Is this the real life?\nIs this just fantasy?"
"As I walk through the valley of the shadow of death"
"Somebody once told me\nThe world is gonna roll me."
"Look\nIf you had\nOne shot\nOr one opportunity."
"Joy to the world\nThe lord is come."
"Carry on, my wayward son\nThere'll be peace when you are done."
"Please allow me to introduce myself."
"At first I was afraid, I was petrified."
"The world was on fire and no one could save me but you."
"Josie's on a vacation far away."
"She's got a smile that it seems to me."
"She was a fast machine, she kept her motor clean."
"Do you have the time to listen to me whine."
```

This is the set of 10 music styles used during evaluation:

```
"Country",
"Electronic",
"Hard Rock",
"Hip-Hop",
"Latin Rock",
"Metal",
"Opera",
"Pop",
"R&B",
"Singer-Songwriter"
```

## A.5 MIDI2AUDIO F0 CONTOURS

Figure 8 shows comparisons of 25 monophonic melodies from the test set. As evident from the plot, the model manages to follow the provided MIDI notes and timing well, while inserting nuances, and varying timbres to the performance.

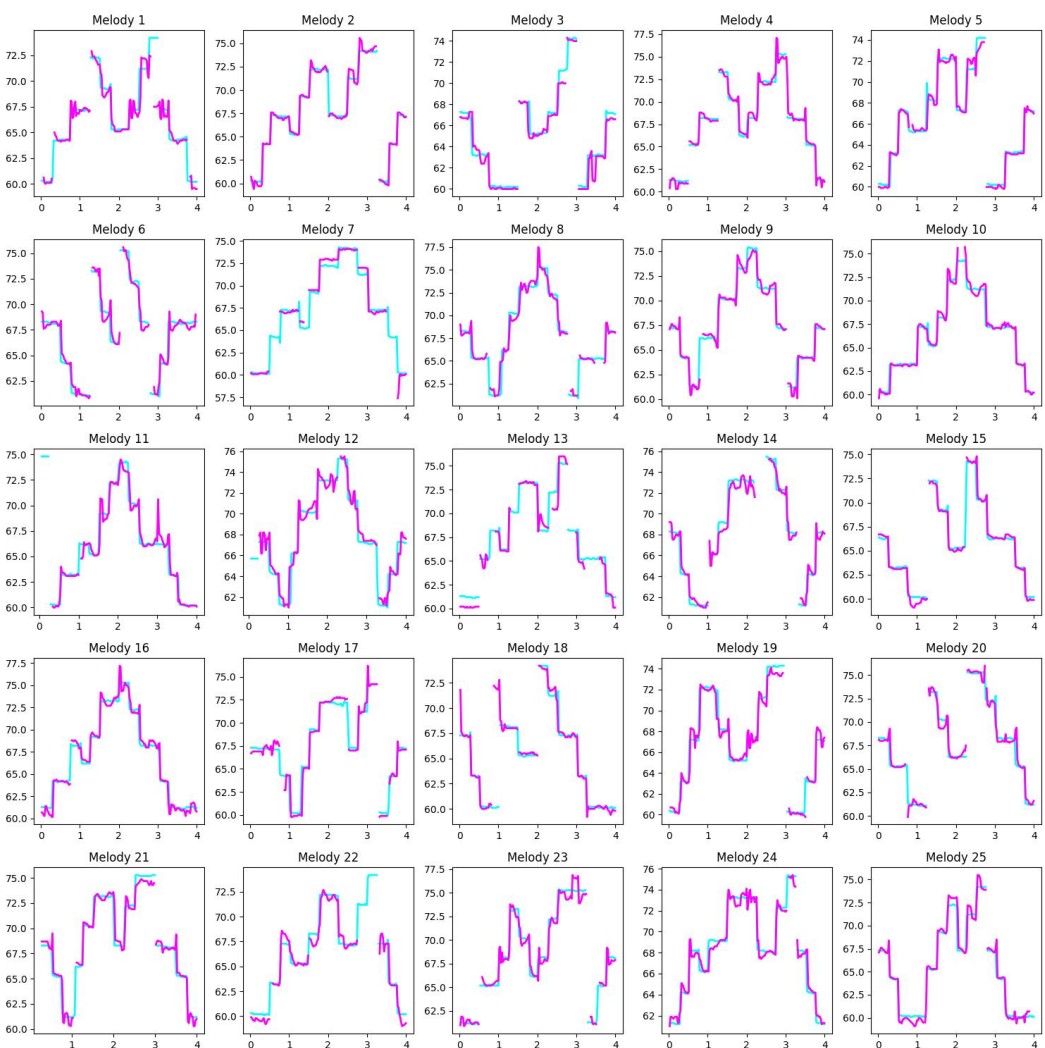

Figure 8: F0 contours of input MIDI (cyan), and generated melodies (magenta). X-axis denotes time in seconds, and Y-axis denotes MIDI pitch.

## A.6 COMPOSITIONALITY

**Attribute Composition (Audio Events):** This is the list of audio events used during Attribute/Event composition in 3.5:

```
 Violin, fiddle; Accelerating, revving, vroom; Water; Acoustic
guitar; Afrobeat; Whistle; Air conditioning; Air horn, truck horn;
Aircraft; Wind
```

