# OpenReview forum: "Fugatto 1: Foundational Generative Audio Transformer Opus 1"
_ICLR.cc/2025/Conference — ICLR 2025 Poster_

### Official Review · Reviewer_FfHR · 2024-10-24

**Soundness:** 3
**Presentation:** 2
**Contribution:** 2
**Rating:** 6
**Confidence:** 5

**Summary:**

This paper introduces Fugatto, a versatile audio synthesis and transformation model capable of following free-form text instructions with optional audio inputs. The paper focuses on enabling generalist audio models with emergent capabilities, similar to large language models (LLMs), but optimized for audio. Fugatto can handle tasks such as audio generation, transformation, and compositional tasks by using a new dataset generation approach and a technique called Composable Audio Representation Transformation (ComposableART).  This method enhances classifier-free guidance, enabling complex operations like combining, interpolating, and negating instructions.

**Strengths:**

(1) Generalist Audio Model: Fugatto offers a broad range of audio generation and transformation capabilities, filling the gap between specialist models and generalist models.

(2) ComposableART: The novel technique extends classifier-free guidance to handle compositional tasks, allowing the model to compose instructions in ways that were not seen during training.

(3) Dataset Generation: The paper provides a strategy for generating diverse and dynamic datasets, using LLMs for instruction creation and data augmentation. The authors claims they will release these the dataset and instruction generation code, which is useful for the research community.

**Weaknesses:**

At this stage, I believe the presentation of this paper is not strong enough. I still have the following questions:

a. How should we understand "unsupervised multitask learning"? In your training tasks, each task has both inputs and labels, such as TTS and TTA. I am not clear on how this relates to unsupervised multitask learning. UniAudio and AudioBox are closely related to your work, and I believe both Fugatto and UniAudio are trained on multiple tasks. Generally, we refer to 'unsupervised learning' when there are no labels, such as in the pre-training of LLMs.

b. Your main framework is based on Flow Matching (FM). How do you control the duration? Since FM is a non-autoregressive model, this could be a challenge. For example, AudioBox or VoiceBox use a phoneme duration predictor. Are you using a duration predictor for TTS? Similarly, for tasks such as singing generation, how is the duration controlled?

c. In Table 1, what qualifies as "large-scale data" in terms of hours? It seems UniAudio uses more than 100,000 hours of audio data. It would be helpful if you could list the number of hours for each model, as this would provide readers with a clearer understanding of what constitutes 'large-scale.'

d. The definition of "Emergent properties": In the context of LLMs, we describe 'emergent properties' as the ability to solve unseen tasks and perform reasoning. However, in this paper, the examples given for emergent properties seem to focus on generating sounds that don’t exist in the real world or the training data. From my perspective, this doesn’t fit the definition of 'emergent properties,' as the model has already learned to understand sound types based on text descriptions. I strongly recommend discussing this point with other reviewers, as it currently seems like a bit of an overstatement.

e. From a high-level perspective, Fugatto follows the multi-task training paradigm used in UniAudio and AudioBox. The authors need to explicitly highlight the advantages Fugatto has over AudioBox. That said, I do agree that the model's performance is impressive.

In conclusion, I agree that building a generalist model is a valuable topic, and this paper demonstrates good performance. However, the authors need to improve the presentation to help readers better understand the contributions.

I am happy to improve my score during the rebuttal stage if the authors solve my concerns.

**Questions:**

Refer to weakness.

---

> ### Author Response · Authors · 2024-11-21
> **Addressing Comments and/or Questions**
>
> Thank you for your thoughtful feedback and for sparking a discussion on "unsupervised multitask learning" and "emergent abilities." We hope that our comments below clarify and align our positions, and hope that you would be willing to raise your score. We are happy to address any other questions you might have.
>
> **a. How should we understand "unsupervised multitask learning"?**
>
> Thank you for this comment, as it seems to both have captured our formulation, as well as a gap in our presentation of it, which we will improve upon in the new manuscript.
>
> Our use of the term “Unsupervised Multitask Learning” follows "Language Models are Unsupervised Multitask Learners" (2019), where p(output_text | input_text) is considered unsupervised and p(output_text | input_text, task_text) is considered supervised. Compared to the text domain, modeling p(output_audio | input_audio) or p(output_audio | input_audio, task_text) does not provide us with the ability to teach the model to follow language based instructions related to sound generation and transformation.
>
> As such, by drawing a parallel with text-LLMs, one can define that p(output_audio | input_audio, instruction, task_text) [template-based instructions] is equivalent to supervised learning, and p(output_audio | input_audio, instruction_text) [free-form instructions] is equivalent to unsupervised. Then, “unsupervised multi-task learning” can be defined as learning to perform multiple tasks with text instructions without "explicitly" providing task conditioning.
>
> We understand that, without this long contextualization, the term “unsupervised multi-task learning” and how it relates to our work may not be clear. As such, we will separately draw the parallel with text-LLMs and find an alternative term to refer to our setup.
>
> **b. Your main framework is based on Flow Matching (FM). How do you control the duration?**
>
> During training for TTS and SVS, we silence-pad the audio up to 20 seconds and compute the loss with weight 0.1 on the padded region, following E3TTS (https://arxiv.org/abs/2311.00945). During inference, we sample a 20-second long noise and can, arguably, control the speech rate through language.
>
> **c. In Table 1, what qualifies as "large-scale data" in terms of hours?**
>
> We will update the paper to list the number of hours used on each model and mark UniAudio as using "large-scale" data. On another note, we will rectify an arithmetic mistake in our total number of hours: 20 million rows with 10 seconds of audio equates to 50,000 hours of audio, not 2.9 million hours.
> Though the on-the-fly augmentations in audio and instructions considerably do increase the number of audio and text pairs to millions,  we prefer to provide the lower bound in number of audio hours.
>
> **d. The definition of "Emergent properties":**
>
> We believe that "emergent properties" are starting to be studied in Audio foundation models and appreciate the reviewer’s willingness to discuss it within this review process. We follow the definition in “Emergent Abilities of Large Language Models” ( https://arxiv.org/abs/2206.07682), which considers “an ability to be emergent if it is not present in smaller models but is present in larger models.”. As we stated in the paper, Fugatto’s smallest model does not possess emergent abilities present in the largest model.
>
> **e. The authors need to explicitly highlight the advantages Fugatto has over AudioBox. That said, I do agree that the model's performance is impressive.**
>
> Please refer to Table 1, the Experiments section comparing Fugatto with AudioBox and UniAudio, and Fugatto's emergent abilities. If these are not sufficient, we kindly ask you to provide suggestions on what aspects you would like to see compared.

---

> > ### Comment · Reviewer_FfHR · 2024-11-24
> >
> > Thank you for your response.
> >
> > My concerns have been partially addressed.
> > For Table 1, I suggest that the authors carefully review it. For instance, AudioBox can be considered as trained on a large-scale dataset. While I agree with the authors’ use of this table to highlight their work, please ensure that the authors of the cited paper would not disagree or be misrepresented if any inaccuracies or incorrect decisions about their work are made.
> >
> > Additionally, please include a discussion section in the final version, particularly addressing the following points:
> >
> > 1. How to determine the duration.
> >
> > 2. The definition of "Emergent Properties."
> >
> > I recommend emphasizing that your interpretation of "Emergent Properties" is specifically limited to task combination. From my perspective, this differs from the "Emergent Properties" observed in LLMs. Please ensure this distinction is clear to avoid confusing readers.
> >
> > I will increase my score to 6.
> >
> > Best wishes.

---

> ### Author Response · Authors · 2024-11-24
> **Thank you**
>
> Thank you for your the short discussion, your comments, and the opportunity they create to improve the paper and, by consequence, the knowledge it shares with our community. We will modify the manuscript to incorporate your suggestions.
>
> And thank you for increasing your score to 6. Though the proposed change hasn't been reflected on OpenReview yet, we assume it will be reflected towards the final score.

---

### Official Review · Reviewer_sTa1 · 2024-10-31

**Soundness:** 3
**Presentation:** 2
**Contribution:** 1
**Rating:** 6
**Confidence:** 4

**Summary:**

The paper proposes Fugatto 1, a versatile audio synthesis and transformation model, has capable of following free-form text instruction with optimal audio inputs. To address the problem that audio data does not inherently contain the instructions that were used to generate it, the paper introduces a specialized dataset generation approach to enhance the data reveals meaningful relationships between audio and language. Additionally, the paper proposes ComposableART with CFG to enhance the compositional abilities of the model.

**Strengths:**

The paper proposes Fugatto, a large foundational audio synthesis and transformation model. This paper attempts to address the problem of audio data does not highly correspond to the text-instruction by a data generation strategy and extend CFG to support compositional guidance. The paper provides extensive experiments across different tasks in the audio domain.

**Weaknesses:**

The paper is a bit difficult to follow, and the proposed methods are trivial incremental. The main contribution of this paper is to train a generalist model with additional enhanced diverse synthesized datasets. I admit that the motivation of exploring a generalist model to benefit downstream tasks is good; however, the paper methodology lacks insights in the domain of audio synthesis and generation. The contribution of this paper for audio domain research is limited. In my opinion, this paper is more suitable for a technical report.

I feel lost when reading the experiment section, the paper should provide at least a brief introduction for evaluation metrics used in the experiments, experimental details for adapting the proposed method in each single-task, and insights about why to adapt the method to each different single-task.

Even though the paper showcased its applicability in various audio related tasks, however, it is not convincing to me the advantage of the proposed generalist model compared to other specialist models for each single-task. For example, in the TTS experiment, only speech similarity and intelligibility been evaluated, a further analysis, such as MOS study or F0 visualization compared to other methods and GT, is necessary for evaluating speech naturalness and quality to showcase the method can provide more natural speech than existing methods; No model comparison for the SVS experiment; Table 3(b) compares its performance with other specialist models, however, MusicGen and AudioLDM2 have different focuses (music v.s., multi-modality audio generation). The experiment is not a fair comparison and not convincing.

**Questions:**

1. If I understand correctly, the paper provides an incremental method for building a large foundational audio generation model. What is the technical contribution of this paper?

2. Have you verified the quality of the synthesized new dataset? How do you ensure the synthesized data is high-quality and strongly corresponds to the prompt instruction?

3. Will you release the codes, the new datasets and the pretrained model?

---

> ### Author Response · Authors · 2024-11-21
> **Addressing Comments and/or Questions**
>
> Thank you for your feedback. Before addressing your comments below, we emphasize that Fugatto is a generalist model that is never adapted to a single-task. We hope that the comments clarify our paper and hope that you would be willing to raise your score.
>
> **Comment and/or Question 1: the proposed methods are trivial incremental.**
>
> We acknowledge your feedback regarding the clarity of the paper and will revise the manuscript to improve its clarity.
>
> We appreciate your perspective but respectfully suggest that, given triangle inequality and Table 1 in our paper, characterizing our work as “trivial incremental” not only devalues the significant effort and innovation behind it, but also inadvertently undermines the contributions of other models we compare with. We kindly ask you to read our technical contributions in our general comment, as well as revisit our Introduction section, to re-asses your characterization of our work and other works as trivial.
>
> **Comment and/or Question 2: I admit that the motivation of exploring a generalist model to benefit downstream tasks is good;**
>
> We appreciate your acknowledgment of the value that generalist models can bring to downstream tasks. We emphasize, however, that Fugatto is a generalist model that is comparable or better than specialist models. We also highlight that our work focuses on the unique properties and benefits of generalist models, especially emergent properties.
>
> **Comment and/or Question 3: The paper methodology lacks insights in the domain of audio synthesis and generation.**
>
> We appreciate your perspective but respectfully disagree with the characterization of our work as lacking insights in the domain of audio synthesis and transformation. We believe this assessment overlooks our contributions, as well as the positive feedback we have received from other reviewers. Please see our technical contributions in our general comment.
>
> **Comment and/or Question 4: experimental details for adapting the proposed method in each single-task**
>
> We emphasize that Fugatto is not adapted for each single-task (TTS, SVS, TTA, etc…). Fugato is a single generalist model trained to follow instructions expressed in text and optional audio inputs. We appreciate your suggestion and will provide a brief introduction for the evaluation metrics.
>
> **Comment and/or Question 5: For example, in the TTS experiment, only speech similarity and intelligibility been evaluated.**
>
> The evaluation used in our paper for TTS is present in several SOTA TTS models since Vall-E, including P-Flow, VoiceBox, and AudioBox.
>
> Nonetheless, we address your request and run SQUIM-MOS, i.e. model based MOS, evaluations to compare Fugatto with samples available on Vall-E’s and UniAudio’s demo page. Our results below further promote our results in the paper, and show that Fugatto surpasses Vall-E and UniAudio in terms of SQUIM-MOS.
>
> SQUIM-MOS Fugatto 4.75  Vall-E      4.00
>
> SQUIM-MOS Fugatto 4.58  UniAudio 3.933
>
> **Comment and/or Question 6: No model comparison for the SVS experiment; MusicGen and AudioLDM2 have different focuses (music v.s., multi-modality audio generation).**
>
> A direct comparison with other SVS model is not possible because, unlike TTS models,  they do not provide the speech prompts used during inference. As such, we do not compare with them not to draw unclear conclusions.
>
> Regarding TTA comparisons with MusicGen and AudioLDM2, we re-emphasize that Fugatto is a generalist model.
>
> **Comment and/or Question 7: What is the technical contribution of this paper?**
>
> Please refer to our technical contributions in our general comment.
>
> **Comment and/or Question 8: Have you verified the quality of the synthesized new dataset?**
>
> Yes, quality and efficacy have been confirmed in many ways:
> 1) All our results were obtained with Fugatto trained with our data generation strategy, and Fugatto is on par or superior to the state-of-the-art.
> 2) Some of the synthetic captions used in Fugatto were confirmed to work on several papers, including https://arxiv.org/abd/2406.15487 and https://arxiv.org/abs/2407.04416v3.
> 3) The LLM-generated instructions were manually inspected with a 1 in 100 test, where we sample 100 instructions and check if at most 1 instruction needs improvement, else we improve the instruction generator by hand or by re-prompting the LLM.
>
>
> **Comment and/or Question 8: Will you release the codes, the new datasets and the pretrained model?**
>
> Yes. We believe that releasing training and inference code, including instruction generators, data and task processors, including a list of datasets, and instructions on how to create new datasets from existing datasets is very important. By releasing such assets, we hope that we will accelerate our journey, as a community, towards a future where unsupervised multitask learning in audio synthesis and transformation emerges from data and model scale. We plan to release checkpoints once the appropriate guardrails to prevent misuse are in place.

---

> > ### Author Response · Authors · 2024-11-24
> > **Request to review the rebuttal**
> >
> > Thank you for taking the time to review our paper. We have addressed your concerns in our submitted response. As the rebuttal period is nearing its conclusion, we kindly request you to review our rebuttal and share any additional comments or concerns you may have. Thank you once again for your valuable feedback and we would be happy to answer additional questions!

---

> > ### Comment · Reviewer_sTa1 · 2024-11-24
> >
> > Thank the author for the clarification, my concerns have been partially addressed. I'm happy to raise my score from 3 to 5. The paper could be further improved by involving domain insights of audio generation.

---

> > > ### Author Response · Authors · 2024-11-27
> > > **Clarifications and Follow-Up on Audio Generation Domain Insights**
> > >
> > > Dear reviewer,
> > >
> > > Thank you for your feedback and for raising your score. Could you kindly elaborate on the specific domain insights of audio generation that you feel are missing from our paper? While we believe our contributions provide substantial insights, we would be happy to incorporate additional domain-specific perspectives to further enhance the work.
> > >
> > > Please let us know what you would like to see more of.

---

> ### Comment · Reviewer_sTa1 · 2024-11-28
>
> This paper lacks technical contribution to the domain of audio research. Although the paper presents an ambitious motivation; however, either using language prompts to build additional dataset for training the model and adjusting model architecture and scales (e.g., DiT , ComposableART, and sampling schedules) are common solutions for generative tasks.
>
> The main research problem in this paper, if I understand correctly, is to increase the data scale and model scale, further to make the model to be generalist. I think this paper may not be counted as a scientific paper, which mainly doing engineering rather than solving the problem with a clear insight on audio domain. In other words, the proposed solution can be directly applied to any generative tasks, such as image. Audio is a general scope that can be classified into many subdomains, such as music, speech, and sounds. Each subdomain has its different focus, which cannot be solved by simply increasing the model scale and data scale.
>
> I suggest the author to think the problem of how to build a generalist audio model deeper, with insights about the inherent nature of audio. In addition, this paper should be revised, with related work section and, as other reviewers suggest, detail experimental details.
>
> In summary, I think this paper lacks enough technical contribution as a scientific paper.

---

> ### Author Response · Authors · 2024-11-28
> **Focus of different domains**
>
> Thank you for reply.
>
> We will soon provide a response describing the adjustments that have been done such that all subdomains in audio (music, speech, the rest) are addressed in a way that does not compromise each other. Meanwhile, we kindly ask the reviewer to familiarize themselves, if that's not the case, with the paradigm described by Hyung Won Chung in the video below, highlighting that different data regimes require different inductive biases. We believe that an understanding of inductive biases will be important in understanding the explanations we will provide regarding adjustments for each subdomain.
>
> https://youtu.be/orDKvo8h71o?si=85brQxlhPTWcWhcy&t=774

---

> ### Author Response · Authors · 2024-11-28
> **Insights about the inherent nature of audio 1 of 2**
>
> Dear reviewer,
>
> below we showcase design choices and contributions in the paper that highlight audio specific insights. Our main design choice in Fugatto is to develop a general model with weaker modeling assumptions, or inductive priors, in consonance with recent work in [LLMs](https://youtu.be/orDKvo8h71o?si=85brQxlhPTWcWhcy&t=774)
>
> Given the diversity in audio subdomains (speech, music, other sounds, captions, music scores, speech prompts), we focus on weak modeling assumptions that support audio synthesis and transformations through text instructions and optional audio inputs in general. This is in contrast with recent works and specialist models that focus on strong assumptions.
>
> Below we highlight challenges that motivate our design choices and disadvantages of other approaches towards building generalist models:
>
> I) Conditioning on Music Scores, Chords, Melodies
>
> Though a musical score can be represented with text (MuseNet and others), this choice comes with several drawbacks:
> 1) Prohibitively long context, specially for complex orchestrations from the likes of Wagner and Mahler
> 2) Challenging to temporally align each instrument or instrument section
> 3) Challenging to represent melodic curves such as the ones in [Xenakis' Aroura](https://www.youtube.com/watch?v=6HmpDpZWLCw)
>
> Our design choice is to use audio generated from MIDI, and provide it to a single audio encoder shared amongst all task. This provides several advantages:
> 1) Context length does not scale with the number of instruments
> 2) Instruments and instrument sections are temporally aligned by construction
> 3) Melodic curves are directly represented
> 4) The audio encoder is shared between all tasks and audio subdomains, promoting emergent abilities.
>
> Our argument can be extended to melodies and chords. For chords, previous works suggest using a look-up table, which is clearly a brute force approach given that even a instrument like the piano, with only 88 unique notes, can produce as many as 10^26 unique chords.
>
> II) Speech Synthesis
>
> Speaker embedding: Some models promote using a speaker verification model for embedding the speaker information, this comes with several drawbacks:
> 1) The information bottleneck from speaker verification models has been proven to result in worse speaker similarity (p-flow, vall-e, voicebox)
> 2) A separate speaker embedding does not provide an audio embedding that is shared amongst domains, possibly limiting emergent capabilities
>
> Our design choice is to use the output of the shared audio encoder. This provides several advantages:
> 1) Model has full access to the speaker's sample, promoting higher speaker similarity
> 2) Full access to the speaker's previous sample, promoting continuation tasks
> 3) A shared audio encoder likely promotes emergent capabilities
>
> F0 control: some models explicitly conditioning on F0 by providing a separate embedding for F0. This comes with disadvantages:
> 1) A separate F0 encoder does not provide an audio embedding that is shared amongst domains, possibly limiting emergent capabilities
> 2) Drastically complicates the data ingestion pipeline, creating difficulties for scaling the data
> 3) Hard to adapt to polyphonies
>
> Our design choice is to provide control over F0 through text captions or through an F0 contour as audio provided to the audio encoder. This provides several advantages
> 1) F0 can be modified with verbal instructions
> 2) Our data generation strategies promotes absolute and relative instructions (higher, lower, more varied contour, less varied contour, etc...).
> 3) Trivial to go from a F0 contour to audio but arguably hard to go from audio to an F0 contour
> 4) Trivial to adapt to polyphonies
> 5) The audio encoder is shared between all tasks and audio subdomains, promoting emergent abilities.
>
> Our argument can be extended to phoneme durations, which can be easily controlled by asking the model to speak slow, fast, slower or faster.
>
> III) Text representation
>
> Though fixed vector length representations like CLIP and CLAP have been used in TTA models, this design choice comes with several drawbacks:
> 1) Due to their fixed length representation, information needs to be compressed to maximize the alignment between caption and audio, removing information that is not related to the captions
> 2) Representation is not adequate for textual instructions, nor the "text" to be sung or said in SVS and TTS tasks
> 3) Such representations are normally interpreted as bag of words, providing very [limited understanding of language](https://www.youtube.com/watch?v=BnpB3GrpsfM)
>
> Our design choice is to instruct the model with a byT5 representation. This has several advantages:
> 1) Supports graphemes, IPA, and characters from non-english languages
> 2) byT5 is trained on vasts amounts of language and, hence possessing a much better understanding of language than CLIP or CLAP.
> 3) Supports captions, instructions, text, html, etc.. all in a single model and shared space

---

> ### Author Response · Authors · 2024-11-28
> **Insights about the inherent nature of audio 2 of 2**
>
> We hope that you agree that the simplicity in our formulation, our ability to easily scale up the data and number of tasks, and the performance of our model reflect our thorough understanding of audio, and deep consideration of the challenges in combining tasks and audio data from different domains, not a lack of insight.
>
> Please note that even though we do not explicitly provide a related works section, it is merged with the introduction given the limit in number of pages and the amount of information we must cover in a work like Fugatto.
>
> We are happy to highlight more challenges and how they were addressed in our paper.

---

> > ### Comment · Reviewer_sTa1 · 2024-11-28
> >
> > Thank the author for this clarification. Now I’m convinced and adjust the score to 6 accordingly. However, the paper should be majorly revised to highlight above as well as a comprehensive related work.  Good luck !

---

### Official Review · Reviewer_xcth · 2024-11-05

**Soundness:** 3
**Presentation:** 3
**Contribution:** 3
**Rating:** 8
**Confidence:** 4

**Summary:**

The paper presents Fugatto, a generalist audio synthesis model capable of handling various audio generation and transformation tasks using text prompts. It introduces a dataset generation method that broadens Fugatto’s capabilities. In addition, ComposableART is proposed which extends classifier-free guidance to allow compositional control over audio generation. The model is evaluated on tasks across different audio domains, showcasing its versatility and competitive performance.

**Strengths:**

1. Fugatto handles multiple tasks with text and audio inputs.
2. The approach to synthetic instruction generation and data enrichment is well-structured, supporting the model’s generalization across diverse tasks
3. ComposableART enables flexible audio composition, interpolation, and negation of prompts, adding control over generated outputs.

**Weaknesses:**

1. Techniques such as using LLMs for synthetic instruction generation lack novelty, which may challenge the paper's originality and scientific contribution.

3. The comparison with state-of-the-art specialist models is limited for certain tasks, and the overall impact of ComposableART on performance remains unclear.

**Questions:**

1. How does ComposableART impact model performance quantitatively on specific tasks? It is recommended that the authors include more subjective and objective comparisons of the proposed system against task-specific models, such as LASS-Net and AudioSep for text-based audio removal.

2. Could the authors clarify which benchmarks or metrics were used to evaluate compositional tasks?

3. As a scientific conference submission, I would personally advise against the use of excessive fancy fonts and varied colors within the paper.

---

> ### Author Response · Authors · 2024-11-21
> **Addressing Comments and/or Questions.**
>
> Thank you for your feedback. Below we attempt to address your questions and commentaries, in hopes that you would be willing to raise your score.
>
> **Comment and/or Question 1: ”Techniques such as using LLMs for synthetic instruction generation lack novelty, which may challenge the paper's originality and scientific contribution.”**
>
> We agree that using LLMs for synthetic instruction generation, as explored in our dataset generation pillars I, III, and V, have been addressed in previous work. With the paragraph below, we hope that you agree that our approach is novel in multiple ways.
>
> First, we unify synthetic instruction generation and dataset creation into a cohesive framework. Furthermore, dataset generation pillars II and IV stand out as innovative contributions: (II) generating absolute and relative instructions enables nuanced tasks like “increase the happiness of this voice,” or “increase the reverb”, while (IV) transmuting datasets uncovers latent relationships, allowing the creation of entirely new tasks such as MIDI2AUDIO and Speech Modulation (Emotion Conversion, Emotion Modulation, Sentence Variation). *Importantly*, we not only propose a dataset and instruction generation strategy, but also demonstrate its effectiveness—our results show that Fugatto achieves performance at least comparable to state-of-the-art specialist models, highlighting the originality and practical impact of our approach.
>
> **Comment and/or Question 2: The comparison with state-of-the-art specialist models is limited for certain tasks. It is recommended that the authors include more subjective and objective comparisons of the proposed system against task-specific models, such as LASS-Net and AudioSep for text-based audio removal.**
>
> We appreciate your feedback and agree that additional evaluations on specific tasks could further demonstrate Fugatto’s capabilities. However, we hope the reviewer recognizes that the existing experiments already provide a broad and comprehensive evaluation across diverse tasks, including TTS, TTA, SVS, Speech Denoising, Upsampling, MIDI2Audio, and Speech Modulation. Furthermore, we hope that you agree that the extensive qualitative examples on our Demo Page, including emergent sounds and tasks, strongly substantiate Fugatto's abilities, versatility and applicability to a large number of scenarios.
>
>
> **Comment and/or Question 3: How does ComposableART impact model performance quantitatively on specific tasks?**
>
> Instead of using ComposableART to improve Fugatto’s performance on benchmarks, which is already comparable or superior to specialist models and UniAudio, our focus with ComposableART is to control sound generation in a new way, to create novel combinations of sounds that don't exist in the training data, and to provide users with the ability to control the influence of each instruction over time.
>
>
> **Comment and/or Question 4: Could the authors clarify which benchmarks or metrics were used to evaluate compositional tasks?**
>
> We used cosine similarity between CLAP embeddings between A and B, where A is obtained from the text description used to create the audio event and B is obtained from the model’s audio output generated with the text description.
>
> **Comment and/or Question 5: As a scientific conference submission, I would personally advise against the use of excessive fancy fonts and varied colors within the paper.**
>
> We appreciate your suggestion and will find an alternative to articulate the entities without using color.

---

> > ### Comment · Reviewer_xcth · 2024-11-23
> > **Response to authors**
> >
> > Thank you for your response. My concerns have been partially addressed.
> >
> > While I acknowledge that the approach to synthetic instruction generation and training with large-scale data is well-structured and represents a solid contribution, the underlying novelty appears limited, as it seems more like an engineering optimization rather than a scientific advancement.
> >
> > The ComposableART component also seems to be an interesting contribution; however, the authors do not provide enough results, analysis, and comparisons to make the contribution distinct.
> >
> > Therefore, I will maintain my original score.

---

> > ### Comment · Reviewer_xcth · 2024-11-28
> >
> > After reviewing the authors' responses to the other reviewers, I have reconsidered my evaluation. I now believe that a generalized paradigm for unified and controllable audio generation represents a significant contribution to the field. I am happy to raise my score to 8. I also encourage the authors to include additional discussion of related work and to emphasize their design choices, particularly in response to reviewer sTa1's feedback.

---

### Author Response · Authors · 2024-11-21
**General Comment**

We kindly thank the reviewers for their comments and suggestions.

Generally speaking, we would like to emphasize that Fugatto is a generalist model that is comparable or better than specialist models, even though it was never adapted to specific tasks. In addition to the list of contributions already provided in our Introduction, below we provide a list of our Technical Contributions and Insights in hopes that this birds-eye view further clarifies our contributions.

In addition, since our submission, we have found new emergent capabilities (New Tasks) from Fugatto related to combining tasks seen during training to perform a new task. We updated our website to illustrate them and provide a concise commentary as well.

On a final note, we observed an arithmetic mistake when computing the total number of hours we used to train our model: 20 million rows with 10 seconds of audio equates to 50,000 hours of audio, not 2.9 million hours as we have reported. We have rectified this mistake and have marked UniAudio as using large-scale data on Table 1.

**Technical Contributions (Ablations):**
1) **t-Sampling Schedules**: We demonstrate that uniform t-sampling is effective across all tasks, while logistic t-sampling, proposed by Stable Audio, significantly degrades TTS performance, highlighting a key insight for generalist models.
2) **Model and data scale**: Our results show that increasing model parameters improves validation losses and delays overfitting. Consistent with "Language Models are Unsupervised Multitask Learners" (2019), we observe that smaller models lack emergent abilities.
3) **“DiT” implementation improvements**: We improve the “DiT” implementation by computing adaptive layer norm in FP32, using GELU as approximate tanh, and initializing final layers to output zero, aligning with our scaled mel-distribution.

**Technical Contributions (Dataset Generation):**
Given that data scarcity is a problem that must be addressed to train generalist state-of-the-art models for audio generation and transformation – audio data does not come with the instructions used to create it – we make the following contributions:
1) We propose a thorough strategy for producing datasets and instruction sets, aiming to address data scarcity and provide a framework that benefits the broader research community. This strategy unifies synthetic instruction generation and dataset creation into a cohesive framework.
2) We propose a dataset generation strategy based on five pillars. While pillars I, III, and V, have been addressed in previous work, pillars II and IV stand out as innovative contributions: (II) generating absolute and relative instructions enables nuanced tasks like “increase the happiness of this voice,” or “increase the reverb”, while (IV) transmuting datasets uncovers latent relationships to create entirely new tasks like MIDI2AUDIO and Speech Modulation.
3) Importantly, we not only propose a dataset generation strategy but also demonstrate its effectiveness—our results show that Fugatto achieves performance comparable to or exceeding state-of-the-art specialist models, underscoring the originality and practical impact of our approach.


**Technical Contributions (Compositionality):**
Recognizing the challenges of achieving compositional abilities (combination, interpolation, negation) from data alone, we introduce ComposableART, an inference method for composing instructions through latent space manipulation, even across different models.


**New Emergent Abilities (New Tasks):**
We were able to find new emergent abilities in Fugatto that showcase its ability to execute a new task that we interpret as a combination of related tasks seen during training. We provide samples in the Emergent Tasks section in our demo page https://fugatto.github.io
1) Fugato is trained on A) TTS with speech prompt and B) Singing Voice Synthesis with a text prompt, but never on A+B.
Surprisingly, we find that the model is able to perform A+B: Singing Voice Synthesis with a speech prompt.
2) Fugato is trained on A) TTA with captions and B) Real Audio from MIDI audio with music styles (MIDI2AUDIO) but never on A+B.
Nonetheless, we find that the model is able to perform A+B: TTA with captions and a melody provided as audio prompt, and the model’s outputs follow the notes in the melody.
3) Fugato is trained on A) MIDI2AUDIO and B) SVS with a text prompt but never on A+B.
We observe that Fugatto can perform A+B: SVS with a text prompt and a melody provided as an audio prompt, and the model’s output follows the notes in the melody.

---

### Meta-Review · Area_Chair_4BtT · 2024-12-20

**Metareview:**

> This paper introduces Fugatto, a versatile audio synthesis and transformation model capable of following free-form text instructions with optional audio inputs. The paper focuses on enabling generalist audio models with emergent capabilities, similar to large language models (LLMs), but optimized for audio. Fugatto can handle tasks such as audio generation, transformation, and compositional tasks by using a new dataset generation approach and a technique called Composable Audio Representation Transformation (ComposableART). This method enhances classifier-free guidance, enabling complex operations like combining, interpolating, and negating instructions.

The paper is of sufficient scope, experimentally well validated against the adequate concurrent models and baselines. The work overall is a positive contribution to the ML research community.

Thanks to reviewers for engaging with the authors during the rebuttal. The paper post-rebuttal meets the criteria for publication at ICLR.

**Additional Comments On Reviewer Discussion:**

Thanks to reviewers for engaging with the authors during the rebuttal. The paper post-rebuttal meets the criteria for publication at ICLR.

---

### Decision · Program_Chairs · 2025-01-22

Accept (Poster)